# Acoustic and language-specific sources for phonemic abstraction from speech

Anna Mai [1] ✉, Stephanie Riès [2,3], Sharona Ben-Haim[4], Jerry J. Shih [5] &
Timothy Q. Gentner [6,7,8]

Spoken language comprehension requires abstraction of linguistic information from speech, but the interaction between auditory and linguistic processing of speech remains poorly understood. Here, we investigate the nature of this abstraction using neural responses recorded intracranially while participants listened to conversational English speech. Capitalizing on multiple, language-specific patterns where phonological and acoustic information diverge, we demonstrate the causal efficacy of the phoneme as a unit of analysis and dissociate the unique contributions of phonemic and spectrographic information to neural responses. Quantitive higher-order response models also reveal that unique contributions of phonological information are carried in the covariance structure of the stimulus-response relationship. This suggests that linguistic abstraction is shaped by neurobiological mechanisms that involve integration across multiple spectro-temporal features and prior phonological information. These results link speech acoustics to phonology and morphosyntax, substantiating predictions about abstractness in linguistic theory and providing evidence for the acoustic features that support that abstraction.

While much is known about the brain's response to the acoustic properties of speech, comparatively less is known about how the neural response to speech sounds integrates into the larger picture of language processing and comprehension. Phonology, as the analytical level in linguistic theory that bridges speech acoustics and morphosyntax, is likely critical to understanding this transformation.

However, many results characterizing phonological processing appear reiterant of purely acoustic results. For example, tuning to spectrotemporal features of speech is spatially organized in both primary and secondary auditory areas[1–5], and sensitivity to phonological features recapitulates that spatial organization[6]. Faced with such similarities, some have questioned whether phonology and its attendant level of abstraction are relevant to speech processing at all[7–10].

Uncertainty around the neurocognitive reality of phonology is likely driven by the fact that phonological information is in general highly redundant with acoustic information[11,12]. Consistent with this, word recognition models fit with purely acoustic features can perform with human-like accuracy[13], and purely acoustic neural encoding models can perform identically to those given phonological information[10]. Such results underscore the richness of the information available purely from speech acoustics.

Nevertheless, divergences between acoustic similarity and phonological similarity are well-established in linguistics[14–16], and some evidence suggests these information types are neurally dissociable as well. For example, Di Liberto et al.[17] show that models containing both continuous spectrotemporal features and categorical phonological

[1]University of California, San Diego, Linguistics, 9500 Gilman Dr., La Jolla, CA 92093, USA. [2]San Diego State University, School of Speech, Language, and Hearing Sciences, 5500 Campanile Drive, San Diego, CA 92182, USA. [3]San Diego State University, Center for Clinical and Cognitive Sciences, 5500 Campanile Drive, San Diego, CA 92182, USA. [4]University of California, San Diego, Neurological Surgery, 9500 Gilman Dr., La Jolla, CA 92093, USA. [5]University of California, San Diego, Neurosciences, 9500 Gilman Dr., La Jolla, CA 92093, USA. [6]University of California, San Diego, Psychology, 9500 Gilman Dr., La Jolla, CA 92093, USA. [7]University of California, San Diego, Neurobiology, 9500 Gilman Dr., La Jolla, CA 92093, USA. [8]University of California, San Diego, Kavli Institute for Brain and Mind, 9500 Gilman Dr., La Jolla, CA 92093, USA. ✉e-mail: acmai@ucsd.edu

features outperform models that contain only spectrotemporal or only phonological features, suggesting that the information accounted for by each of these feature sets is not fully identical. However, the relationship between these two types of features remains to be understood.

Rather than undermining the status of phonology in language processing altogether, we argue that the high degree of overlap between acoustic and phonological information accentuates the importance of careful experimental design to assess the unique contribution of phonological information and its relationship to speech acoustics. Only through investigation at the edges of this informational overlap, at points of acoustic-phonemic divergence, will it be possible to identify neural signatures of purely phonological processes and determine the nature of links between sensory processing and language cognition.

Points of acoustic-phonemic divergence are common cross-linguistically, but are specific to particular languages. That is, while two different languages may have many points of acoustic-phonemic divergence and may share some points of divergence in common, altogether each language has a unique set. In this way, acoustic-phonemic divergences provide a window on linguistic abstraction, because they require higher-order knowledge of a particular language (e.g., Tagalog vs. Quechua).

This study focuses on the phonological opposition between the English abstract categories /d/ and /t/ and their contextual acoustic neutralization to the sound [ɾ], a coronal tap. English phonemes /d/ and /t/ have many acoustically distinct acoustic realizations that are conditioned by the surrounding phonological context. These contextually conditioned variants are called *allophones* of /d/ and /t/. Some allophones of /d/ and /t/ are acoustically distinct from one another, but when either /d/ or /t/ occurs following a stressed syllable and between two vowels, their acoustic contrast is neutralized, and both are pronounced as a coronal tap (e.g., *writing, riding*). This acoustic neutralization is demonstrated in Supplementary Fig. 1 for the stimuli used in this study.

Given that many aspects of auditory processing proceed in a feedforward manner from spectrotemporal features of the sensory input, it is anticipated that many speech responsive sites will demonstrate an acoustic 'surface response', where the responses for all taps are more similar to one another than to the voiceless coronal stop allophone of /t/. However, if it is also the case that phonological context is used to compute phonemic identity during language processing, even when the acoustic contrast between two phonemes is neutralized, then there also should exist sites demonstrating a phonemic 'underlying response', where the neural response to underlyingly /t/ taps (e.g., *writing*) is more similar to the response to other allophones of /t/ (e.g., voiceless alveolar stops) than to underlyingly /d/ taps (e.g., *riding*).

Observing both sites with acoustic surface responses and sites with phonemic underlying responses would provide evidence for a specific kind of phonological abstraction from surface acoustics that to date has only been argued to exist in theory. In this way, points of acoustic-phonemic divergence act as powerful handles on the relationship between sensory and linguistic processing, in this case, providing a window on a one-to-many mapping of acoustic forms to phonological constructs.

Moreover, just as points of acoustic-phonemic divergence elucidate the nature of abstraction from surface acoustics to underlying phonological structure, points of phonemic-morphemic divergence, where morphemes are not in a one-to-one relationship with phonemes, provide critical testing grounds for understanding the junction between phonology and morphosyntax. In particular, many morphemes have more than one distinct phonemic form, where the form that is pronounced is conditioned by the surrounding phonological context. These contextually conditioned variants of morphemes, called *allomorphs*, provide a window on many-to-one mappings from phonological forms to morphological meanings.

The morpho-phonological processes considered in this study are the formation of the English regular past tense and regular plural. The morphemes for the regular past tense and regular plural both exhibit variation in their realization depending on the phonological context. The regular past tense takes one of three forms: a syllabic voiced coronal stop [əd] following [t] or [d] (e.g., *gifted, folded*), a voiceless coronal stop [t] following the remaining voiceless consonants (e.g., *kissed, plucked*), or a voiced coronal stop [d] following the remaining voiced sounds (e.g., *hugged, shoved*). The regular plural analogously manifests in one of three forms: a syllabic voiced coronal sibilant [əz] following sibilants [s, z, tʃ, dʒ, ʃ, ʒ] (e.g., *beaches, palaces*), a voiceless coronal sibilant [s] following the remaining voiceless consonants (e.g., *forests, peaks*), or a voiced coronal sibilant [z] following the remaining voiced sounds (e.g., *mountains, hovels*).

Since similarity of acoustic form and neural response is a well-established principle of auditory processing (e.g.,[6]), it is anticipated that some speech responsive sites will demonstrate a 'surface response', where responses for [z] forms of the plural and [d] forms of the past tense are more similar to word-final non-plural [z] and non-past [d], respectively, than to the voiceless forms of the plural and past tense, respectively. Additionally, if it is the case that morphological identity is abstracted from phonological context, then there should also exist sites demonstrating a morphological 'underlying response', where the neural responses to both voiced and voiceless forms of the plural and past tense pattern together to the exclusion of word final non-plural [z] and non-past [d], respectively.

In this way, observing the distribution of surface and underlying sites for the plural and past tense comparisons has the potential to provide critical evidence for the mental reality of phonemes and morphemes and substantiate basic assumptions of generative linguistic theory.

In addition to these phonologically and morphologically motivated comparisons, this study also makes use of a receptive field estimation technique as a linking hypothesis. Receptive field estimation approaches use mathematically explicit hypotheses about the nature of the relationship between stimulus and response to estimate what features of the stimulus drive a response. Because the effects of hypotheses used by different approaches can be compared straightforwardly through their explicit mathematical definitions, these approaches are immanently well-suited to answer questions concerning the nature of the link between acoustic signals and language representation in the brain.

Receptive field estimation methods can be roughly divided into *linear methods* that correlate the neural response directly to components of the stimulus ($s_i$) and *quadratic methods* that additionally correlate the neural response to pairwise ($s_i s_j$) (or even higher order) products of stimulus components as well. Linear methods include the spike-triggered average (STA)[18], maximally informative dimensions (MID)[19], and first order maximum noise entropy models (MNE)[20]. Quadratic methods include the spike-triggered covariance (STC)[21] and second-order MNE models[20].

This study compares the relative abilities of receptive field components recovered by first-order and second-order MNE models to reconstruct the neural response to speech. In doing so, this study assesses whether the neural response to speech is impacted by the stimulus covariance. To date, MNE models have been used to characterize the receptive fields of putative single neurons based on spiking activity in visual and auditory areas of nonhuman animals[22–25]. Here, we build on those successes, using MNE models to reconstruct receptive fields from human intracranial LFP activity and assess whether phonemic category information aids in the reconstruction of higher-order sensory receptive fields.

In this study, a combination of MNE models, standard mixed effects models, and targeted linguistic comparisons are used to show how neural activity integrating over both acoustic and phonological information supports linguistic abstraction. Language-specific patterns of acoustic-phonemic divergence are leveraged to demonstrate the *causal efficacy*[12] of the phoneme as a unit of analysis and to distinguish its contributions from those of spectrotemporal cues. Mixed effects models confirm the applicability of this finding to contexts beyond acoustic-phonemic divergence and show how the contribution of phonological information to the neural response varies across frequency bands. Finally, using MNE models applied to human LFP data, this work isolates features of the stimulus responsible for the efficacy of the phoneme, providing evidence for both neural features and stimulus features that support phonology-specific activity in the brain. In total, these results provide clear substantiation of predictions about the nature of abstractness in phonology and evince acoustic features that support that abstraction.

## Results

### Acoustics, phonology, and morphology drive neural activity

For assessing acoustic-phonemic divergence with coronal stop neutralization, acoustic sites and phonemic sites were defined using a sliding-window one-way ANOVA with 100ms windows and 50ms overlap. For a site to be considered an acoustic site, there must have existed at least one time window with a significant ANOVA for which a Tukey's post hoc test indicated that there was a significant difference ($\alpha = 0.05$) between surface [t] and tap /t/ tokens and between surface [t] and tap /d/ tokens but no significant difference between tap /t/ and tap /d/ tokens. Alternatively, for a site to be considered a phonemic site, there must have existed at least one time window with a significant ANOVA for which a Tukey's post hoc test indicated that there was a significant difference between tap /d/ and tap /t/ tokens and between tap /d/ and surface [t] tokens but no significant difference between surface [t] and tap /t/ tokens.

Similarly, for assessing phonemic-morphemic divergence with the regular past tense, surface sites were considered to be those where the sliding-window ANOVA was significant for at least one time window and for which a Tukey's post hoc test indicated that there was a significant difference ($\alpha = 0.05$) between past tense [t] and past tense [d] tokens and between past tense [t] and word final non-past [d] tokens but no significant difference between past tense [d] and word final non-past [d] tokens. For assessing phonemic-morphemic divergence with the regular plural, surface sites were defined as those with at least one time window for which a Tukey's post hoc test indicated that there was a significant difference ($\alpha = 0.05$) between plural [s] and plural [z] tokens and between plural [s] and word final non-plural [z] tokens but no significant difference between plural [z] and word final non-plural [z] tokens.

For the two morphological patterns, the surface similarity comparison collapses the distinction between phonological surface and underlying similarity. We take the plural comparison as an example to illustrate this point; the following points apply analogously to the past tense comparison. That is, the phonemic, underlying form of the plural is typically considered to be /z/[26,27]. Similarly, the phonemic underlying form of word-final non-plural [z] sounds is also /z/. The morphological surface similarity response groups plural [z] and non-plural [z] together to the exclusion of plural [s]. In this way, the morphological surface similarity response groups sounds together that are both acoustically similar ([z]) and phonemically similar (/z/). Thus, sites identified as morphological surface similarity sites are not comparable to the acoustic surface sites identified by the tap comparison because morphological surface similarity sites conflate phonological surface and underlying similarity.

Morphological underlying sites were considered to be those for which the comparison of evoked responses to past tense /t/, past tense /d/, and word-final non-past /d/ resulted in at least one time window indicating a significant difference between word-final non-past /d/ and past tense /t/ tokens and between word-final non-past /d/ and past tense /d/ tokens but no significant difference between past tense /t/ and past tense /d/ tokens. Similarly, for the regular plural alternation, morphological underlying sites were those for which there was at least one time window indicating a significant difference between word-final non-plural /z/ and plural /s/ tokens and between word-final non-plural /z/ and plural /z/ tokens but no significant difference between plural /z/ and plural /s/ tokens. In this way, morphological underlying sites were those which maintained language-specific morphological similarity based on the meaning of similar word-final sounds rather than maintaining the comparably less abstract similarity of their acoustic or phonemic realization.

All sites that were speech responsive for at least one band are shown in Fig. 1b-e. Of the roughly 485 speech responsive electrodes for each band, an average of 31 acoustic surface sites (SD ± 22.4) and 190 phonemic underlying sites (SD ± 38.5) were observed for the coronal stop–tap alternation in each band. On average, six electrodes (SD ± 2.4) per band were categorized as both surface and underlying sites for the tap comparison. The number of acoustic and phonemic sites observed across bands is summarized in Table 1, and examples of the response observed at acoustic and phonemic sites are shown in Fig. 2. For the past tense alternation, for each band an average of 73 (SD ± 23.9) were categorized as surface sites, 46 (SD ± 8.8) were categorized as morphological underlying sites, and one site (SD ± 1.1) was categorized as both a surface and morphological site. For the plural alternation, 47 (SD ± 14.2) sites were categorized as surface sites, 45 (SD ± 12.2) were categorized as morphological underlying sites, and one site (SD ± 0.5) was categorized as both a surface and morphological site.

To assess the likelihood of observing these numbers of surface and underlying response sites by chance, an expected null distribution was generated for each frequency band by performing the statistical analysis described above using 1,000 arbitrary pairs of phones (i.e., A, B) with an arbitrary split of one phone (i.e., A, $B_x$, $B_y$). For each arbitrary set of phones, surface sites were identified as those with at least one time window in which there was a significant difference between the evoked response to A phones and the evoked response to B phones, but no significant difference in the evoked response to $B_x$ and $B_y$ phones. Underlying sites were identified as those with at least one time window with no significant difference between A and $B_x$ phones but a significant difference between those phones and $B_y$ phones. In this way, the null distribution was generated from the real, recorded data. Spatially correlated activity is thus preserved in the null distribution, accounting for the possibility that such correlated activity could inflate the number of observed surface and underlying sites.

Compared against these null distributions, the numbers of observed sites exhibiting surface and underlying patterns of activity were greater than would be expected by chance for each band. For example, for the high-gamma band, based on the generated distribution, the probability of observing at least 111 phonemic underlying sites and at least 80 acoustic surface sites for the coronal stop–tap alternation was <0.1%. Figure 3 shows the null distributions of surface and underlying sites, with a dotted gold line marking the bound for 95% of the distribution's mass and blue points indicating the observed number of surface and underlying sites for the coronal tap comparison.

The numbers of sites exhibiting surface and underlying patterns of activity were also greater than would be expected by chance for both the regular past tense and plural comparisons. The probability of observing the past tense pattern of at least 33 surface sites and at least 32 underlying sites was 0.1% for the high-gamma band, and the probability of observing the plural pattern of at least 19 surface sites and at least 38 underlying sites was 0.2% for the same band. The expected and

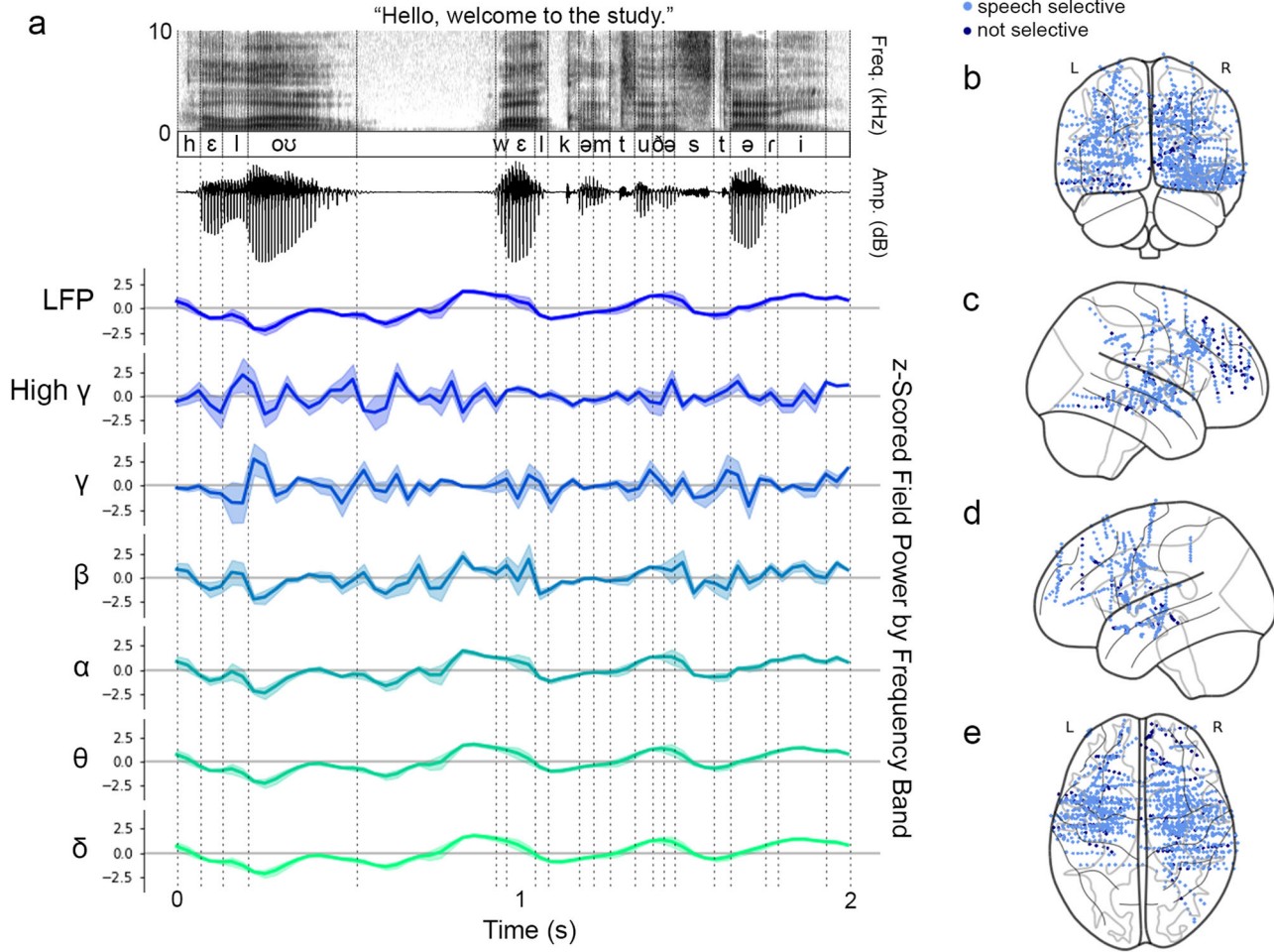

**Fig. 1 | Stimulus-locked activity was recorded from speech responsive sites across multiple lobes. a** Neural activity was filtered into functionally relevant bands and labeled with time-aligned stimulus features, including spectrographic and phonemic label information. **b** Coronal view of coverage for nine patients, warped to standard MNI space. **c** Sagittal view of right hemisphere. **d** Sagittal view of left hemisphere. **e** Axial view. For all views, electrodes that were speech responsive for at least one frequency band are shown in light blue, and non-responsive sites are shown in dark blue. The distribution of speech responsive sites across bands is shown in Supplementary Fig. 2. Subfigures (**b**–**e**) were created using the Python package `nilearn` (DOI: 10.5281/zenodo.8397156).

**Table 1 | Count of sites demonstrating a surface similarity pattern or an underlying similarity pattern for each frequency band**

| Band | Coronal Tap Comparison | | Plural Comparison | | Past Tense Comparison | |
|---|---|---|---|---|---|---|
| | Surface | Underlying | Surface | Underlying | Surface | Underlying |
| δ: (1–3Hz) | 16 | 211 | 59 | 42 | 95 | 55 |
| θ: (4–7Hz) | 16 | 221 | 59 | 40 | 99 | 54 |
| α: (8–12Hz) | 21 | 222 | 51 | 40 | 90 | 50 |
| β: (13–30Hz) | 24 | 198 | 38 | 38 | 61 | 49 |
| γ: (31–50Hz) | 30 | 178 | 54 | 72 | 58 | 36 |
| High-γ: (70–150Hz) | 80 | 111 | 19 | 38 | 33 | 32 |

The observation of more surface sites than underlying sites for the morphological comparison is likely due to the fact that surface similarity for the morphological comparisons is analogous to both surface and underlying similarity at the phonological level.

observed distributions of surface and underlying sites are shown in Fig. 3, with the expected distributions shown as heatmaps, and single red (past) and green (plural) points indicating the observed number of surface and underlying sites for the two morphophonological comparisons.

**Phoneme labels best explain low-frequency band power**
Results from the coronal tap, plural, and past tense comparisons provide evidence for distinct contributions of categorical phonemic knowledge and acoustic processing to the neural response to speech.

Here we propose and assess a basic model of the relationship between these two feature types using linear mixed effects (LME) models similar to those reported by Di Liberto et al.[17]. For each participant, for each of the seven neural response types (=six classic frequency bands and broadband LFP), seven LME models were fit. Each model fit neural response with electrode channel and excerpt speaker as random effects and either only spectrographic features, only phonemic label features, or both spectrographic and phonemic label features as fixed effects. Figure 4 illustrates how these three base models where constructed. Four additional models were also created, in which the

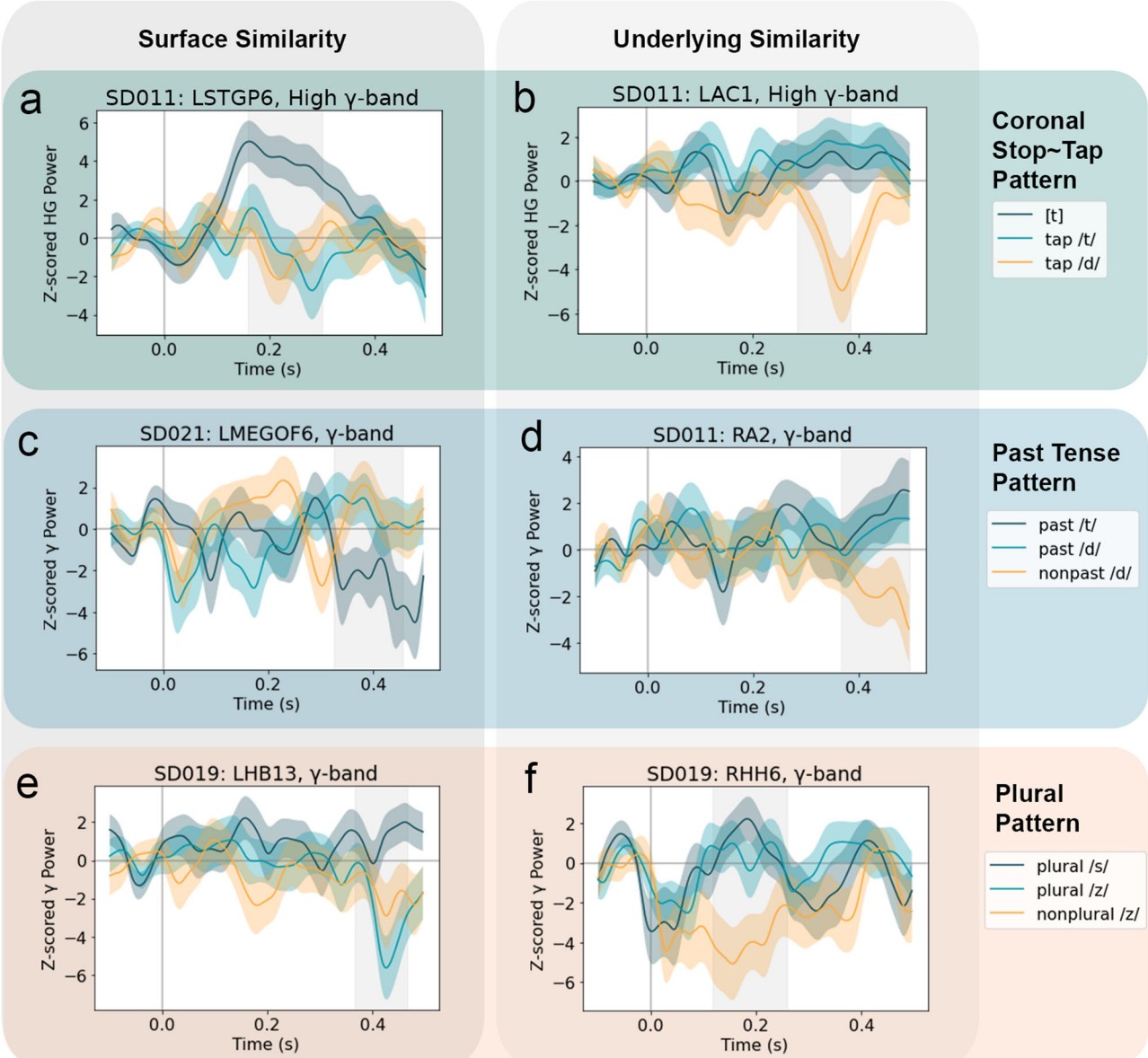

**Fig. 2 | Responses to coronal tap, plural, and past tense comparisons exhibit both surface similarity and underlying similarity patterns.** Subplot titles indicate the subject identity, channel name, and response band being plotted, and each shows the time course of band power z-scored relative to baseline (-100 to 0 ms). (**a**) and (**b**) show sites identified by the coronal stop-tap alternation. The evoked response to tokens of [t] are shown in dark gray (*n*=796); the evoked response to taps derived from /t/ (tap /t/) is shown in teal (*n*=183); and the evoked response to taps derived from /d/ (tap /d/) is shown in gold (*n*=79). (**c**) and (**d**) show sites identified by the past tense alternation. The evoked response to /t/ allomorphs of

the past tense are shown in dark gray (SD021: *n*=15, SD011: *n*=20); the evoked response to /d/ allomorphs of the past tense is shown in teal (SD021: *n*=52, SD011: *n*=53); and the evoked response to word-final non-past tokens of /d/ is shown in gold (SD021: *n*=141, SD011: *n*=138). (**e**) and (**f**) show sites identified by the plural alternation. The evoked response to /s/ allomorphs of the plural is shown in dark gray (*n*=51); the evoked response to /z/ allomorphs of the plural is shown in teal (*n*=105); and the evoked response to word-final non-plural tokens of /z/ is shown in gold (*n*=194). For all subplots, shading indicates ±SEM. The distribution of sites across bands for each comparison is shown in Supplementary Fig. 3.

phonemic or spectrographic features had either been shuffled within each excerpt or shuffled across the full recording session. Models were then compared within participant and neural response type using the Akaike Information Criterion (AIC)[28], and all best-fit models carried 100% of the cumulative model weight and had an AIC score >200 lower than other models.

Power in delta, theta, and alpha bands was best fit by LME models that included both spectrographic features and phonemic labels (s1p1). For these bands, the s1p1 model was the best fit for nine out of the ten participants, and the p1 model was the best fit for one participant (SD012). Power in the beta band was best fit by the s1p1 model for eight participants and the p1 model for two participants (SD010,

SD012). These results suggest that power in these bands is driven in part by phonemic category information that is not reducible to speech acoustics. For power in gamma and high-gamma bands, the best fit model varied across individuals. For gamma power, eight participants' data were best fit by the model that included only spectrographic features (s1), and two participants' data were best fit by the s1p1 model that included both spectrographic and phonemic label features (SD013, SD018). Similarly, for high-gamma power, eight participants' data were best fit by the s1 model that included only spectrographic features, and the remaining participants data were best fit by the s1p1 model that included both spectrographic and phonemic label feature sets (SD011, SD018). These results suggest that power in frequencies

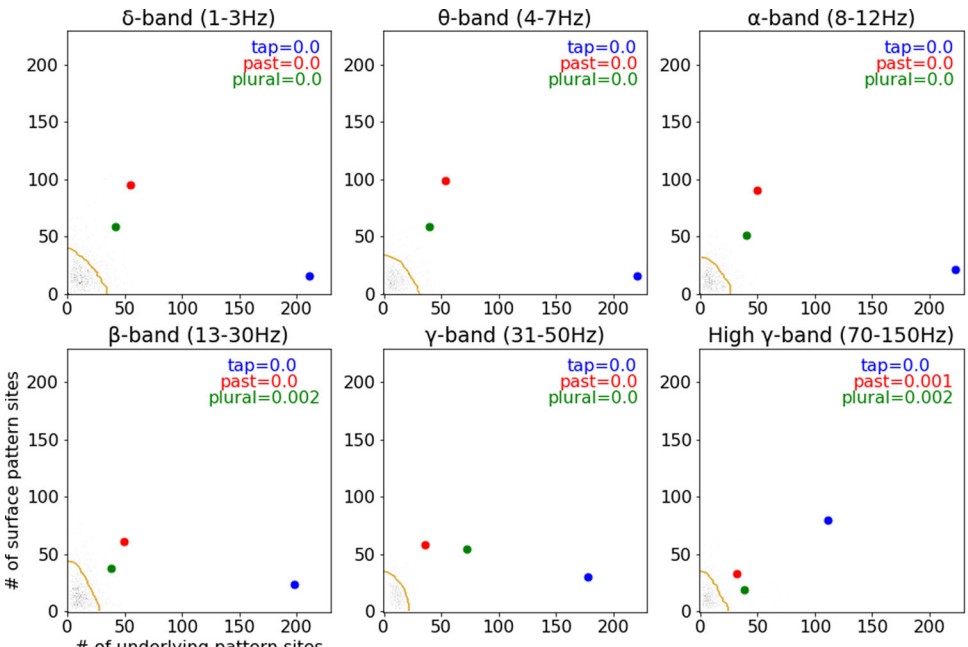

**Fig. 3 | Surface similarity and underlying similarity patterns are not random.** Number of significant sites observed for each (morpho)phonological comparison for each neural response band relative to the generated null distribution for that band. Each null distribution (grays) contains 1000 comparisons. Vertical axes indicate the number of sites selective for surface identity observed for each comparison, while horizontal axes indicate the number of sites selective for underlying identity observed for each comparison. The proportion of the null distribution that contains at least as many surface and underlying sites as were observed for each comparison is indicated in the top right corner of each plot. Values for the tap comparison are blue; values for the past tense comparison are red; and values for the plural comparison are green. Dashed gold lines delimit the boundary containing 95% of the null distribution. Additional detail for each of the three comparisons is given in Supplementary Figs. 4–7.

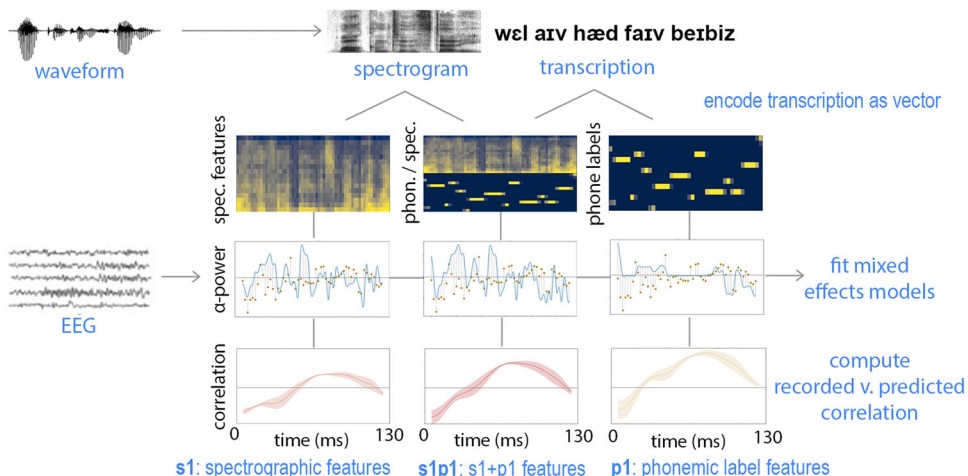

**Fig. 4 | LME approach models neural activity as a combination of spectrographic and phonemic label features.** For each stimulus waveform, spectrograms are computed and time-aligned phonemic-level transcriptions are assigned. Transcriptions are additionally one-hot encoded (top row). Three classes of model are created from these features: a purely spectrographic model (left column), a purely phonemic label model (right column), and a model containing both feature sets (middle column). For each band of neural activity, mixed effects models are fit. Model weights are used to reconstruct a predicted response for each band, and the correlation between the predicted and recorded neural response is calculated.

above 30Hz are primarily driven by speech acoustics rather than phonemic category information.

For broadband LFP, nine participants' data were best fit by the s1p1 model that included both spectrographic features and phonemic labels, and one was best fit by the p1 model (SD012) that contained only phonemic label features. Given the $1/f$ structure of LFP data, this neural measure is dominated by lower frequencies; thus, the fact that the best fit models for lower frequency bands are also the best fit models for the broadband LFP is somewhat expected. What is notable, however, about the match between the best fit models for lower frequency band power and LFP is that LFP and band power are different kinds of neural measures. LFP is a measure of extracellular voltage fluctuation over time, and band power measures are derived from LFP through a series of non-linear transformations: LFP activity is filtered into logarithmically increasing frequency bands and the analytic amplitude of those bands is computed and then averaged across bands. On their own, the LFP results demonstrate that broadband LFP contains phonemic category information that is not reducible to

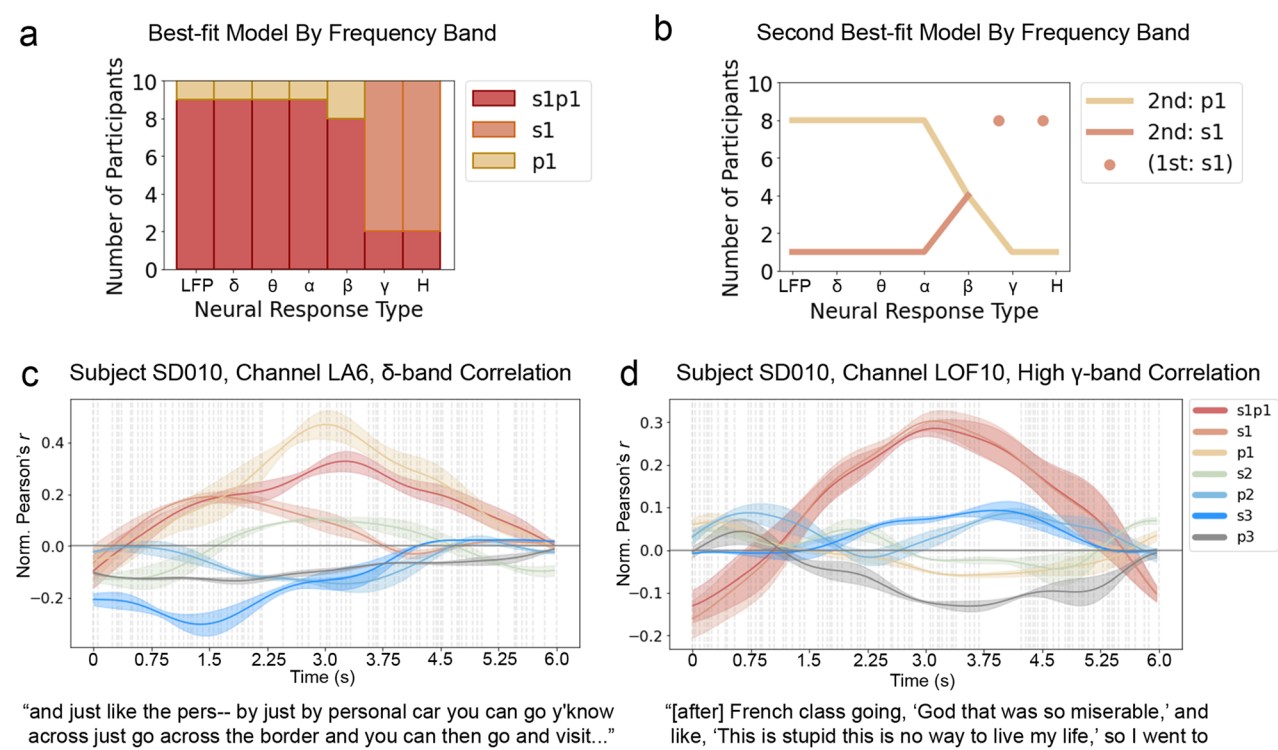

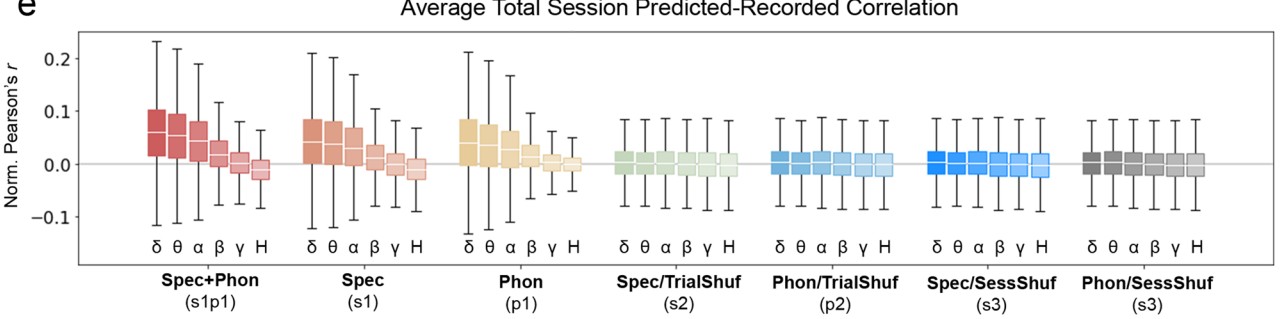

**Fig. 5 | Phonemic information dominates lower frequency band responses to speech. a** For LFP and delta–beta bands, models that include phonemic label information in addition to spectrographic information (s1p1, red) best fit the neural response to speech for most all participants. At higher frequencies, most participants' data are best fit by models containing solely spectrographic features (s1, orange). **b** Models containing phonemic label information alone (p1, yellow) are more likely to provide a better fit for neural response at lower frequencies than at higher frequencies, where models containing spectrographic information alone (s1, orange) are more likely to provide a better fit. **c** Time-varying correlation between predicted and recorded neural response for the delta power response of one electrode, illustrating a period of sustained advantage of the p1 model (yellow) over other models, including the s1p1 (red) model. **d** Time-varying correlation between predicted and recorded neural response for the high-gamma power response of one electrode, illustrating a period of sustained advantage of the s1 model (orange) and s1p1 (red) models over models that do not contain unshuffled spectrographic information. For (**c**) and (**d**), gray vertical lines indicate phoneme boundaries for the transcript provided below the plot. **e** Predicted-Recorded correlations averaged across all speech responsive electrodes for all participants. For each model, box-plots represent n=5566 delta, n=5054 theta, n=6059 alpha, n=5997 beta, n=5869 gamma, and n=3772 high-gamma Pearson's r measurements. White horizontal lines indicate distribution medians, colored boxes indicate interquartile range (IQR), and black whiskers extend a further 1.5 IQR.

speech acoustics. Combined with the results from lower frequency band power, the LFP results further suggest that this information is robust to the set of transformations involved in deriving band power from LFP.

As a model selection tool, the AIC straightforwardly aids the selection of the most probable model from a set of models. More generally, however, the AIC facilitates model ranking, and given the partially nested structure of the models under comparison in this study, the relative ranking of the s1 and p1 models is worthy of assessment. Especially where s1p1 was the best-fit model, assessing the relative fit of s1 and p1 models provides insight concerning which of the two-component feature sets of the s1p1 model contributed the most to its goodness-of-fit. For each subject and response type, the model with the second lowest AIC score was >100 times as

likely as the third ranked model. As shown in Fig. 5b, models containing only phonemic label features were most likely to be the second ranked model for lower frequency bands. However, with increasing band frequency, the s1 model becomes more likely to provide a better fit for the data until, for gamma and high-gamma bands, it is the best fit model overall. These results provide support for the generalization that phonemic labels better explain power at lower frequencies, while acoustic features excel at explaining power in higher frequencies.

**Phoneme identity and acoustic covariance interact**
Nearly all best-fit mixed effects models included spectrographic features, highlighting the pervasive importance of acoustic information in speech processing. However, in lower frequency bands in particular,

categorical phonemic information also played a substantial role, explaining variability in the neural data that was not otherwise explained by spectrographic information. This observation raises two questions: First, which aspects of the stimulus are most important in explaining the neural response to speech? And second, what is the nature of the relationship between spectrographic and phonemic label features? Maximum Noise Entropy (MNE) models provide the analytical tools to address these questions because they allow one to test specific claims about the nature of the link between the stimulus and the neural response. Here, by comparing the fit of linear models to quadratic models, we determine the extent to which relationships between pairs of features impact neural fit, and by comparing the fit of models given phonemic label information to models fit with solely spectrographic information, we determine the role that phonemic information plays relative to speech acoustics.

As shown in Fig. 6, for each channel of neural activity, MNE models estimated numerous receptive fields, including fields that corresponded with an increase in the neural response (Fig. 6b,e) and fields that corresponded with a decrease in the neural response (Fig. 6a,d). And when MNE models were fit with phonemic label information, that information was often visible in the estimated receptive fields, as can be seen by comparing the top row of each receptive field in Fig. 6a, b with those of Fig. 6d, e. From the receptive fields estimated by the MNE models, predicted neural data was generated and compared against the recorded neural data.

To assess whether model order and/or the availability of phonemic label information contributed to the quality of MNE model fits, five mixed effects models were fit. These models predicted the Fisher Z-transformed Pearson correlation between predicted and observed neural responses based on the neural response type, model order, the availability of phonemic label information, the interaction of model order with the availability of label information, and whether the model feature weights were shuffled. Subject and channel were included as random effects. The formulas for these models are given in Supplementary Table 1, and Fig. 7 shows examples and summaries of the correlation data that these models fit. In these models, `model` is a binary feature referring to whether the predicted fit was generated using only the linear MNE feature or predicted using both the linear and quadratic features; the feature `label` indicates whether or not the MNE model was fit using labeled or unlabeled spectrographic features; and the feature `shuffle` indicates whether or not the predicted fit was generated using MNE features whose weights had been shuffled. The model containing only the `shuffle` parameter as a fixed effect was used as a baseline.

Model selection was determined using the AIC. Of the five models, the model containing `model`, `label`, and `model:label` features best fit the data, having an AIC score 54.75 units lower than the next best model and carrying 100% of the cumulative model weight. The next-best fit model included `label` and `shuffle` features and had an AIC score 1.60 units less than the model that additionally included the `model` feature. The baseline model containing only the `shuffle` feature outperformed a model that was minimally augmented by the addition of the `model` feature, garnering an AIC score of 1.59 units lower than the model containing both `model` and `shuffle` features. Together, these results suggest that while MNE model complexity alone does not positively impact prediction quality, its interaction with the label status of the spectrograms used to fit the MNE models is substantial. In other words, when phonemic label information is available, information in the stimulus covariance can be used to more accurately predict neural response. Without phonemic label information, the stimulus covariance does not contribute substantially to model goodness-of-fit.

The failure of the `model` feature to contribute independently to goodness-of-fit was not predicted. Ceteris paribus, the higher dimensionality of the quadratic model is expected to recover a higher proportion of the variability in the neural response than the linear model,

as has been demonstrated by Kozlov and Gentner[22]. However, whereas Kozlov and Gentner[22] recorded from songbird auditory areas only, the data in this study were recorded from speech-responsive electrodes, the majority of which were not localized to auditory areas. Given that the receptive fields in this study were estimated based on primarily spectrographic input features, it may be the case that the higher dimensionality of the quadratic model does not confer a benefit in non-auditory areas where spectrographic information contributes less to neural response. Nevertheless, the fact that the interaction feature (`model:label`) does substantially improve model goodness-of-fit suggests that information about spectrographic covariance is useful for predicting neural response when it is reinforced by categorical phonemic label information. In this sense, the auditory stimulus covariance structure and phonemic identity synergistically impact neural response, jointly providing information available in neither feature set independently.

### Phonemic explanatory power requires language knowledge

Both LME and MNE models of the neural response to speech indicated that categorical phonemic information contributes to the prediction of the neural response to natural speech. Next, we asked whether the utility of phonemic label information in these models requires specific language knowledge. That is, it could be the case that phonemic label information plays a significant role in these models simply because it reinforces acoustic information. If this were the case, we would expect the addition of phonemic label information to improve model fit both when participants were listening to a language they understand and are familiar with (i.e., English) and when listening to a language they do not understand and are unfamiliar with (i.e., Catalan). Alternatively, phonemic label information could play a significant role in these models because it provides language-specific category membership information that is not otherwise available in the speech acoustics. If this were the case, phonemic label information should only contribute significantly towards explaining neural activity while participants are listening to speech in a language that they are familiar with (English) and not while listening to speech in a language they are unfamiliar with and do not understand (Catalan).

To assess whether this was the case, families of mixed effects models were fit on the predicted-recorded correlation values for the predictions of each of the seven LME models and each of the four MNE models.

For the LME models, each model for assessing the role of specific language knowledge predicted the Fisher Z-transformed Pearson correlation between predicted and observed neural responses to particular excerpts on the basis of the LME model type (`model`: s1p1, s1, p1, etc.) and the band of neural activity. A second model had an additional feature (`lang`) for the language being spoken in the excerpt (English or Catalan), and a third model additionally included an interaction term for excerpt language and model type (`lang:model`). For each of these three models, subject and channel were included as random effects. The formulas for the three models in this family are given in Supplementary Table 2.

Model selection was then determined using the AIC. Of the three models, the model containing both `lang` and `lang:model` features in addition to `model` and `band` features best fit the data, having an AIC score 609.59 units lower than the next best model and carrying 100% of the cumulative model weight. The next-best fit model included only `lang`, `model`, and `band` features and had an AIC score 65,120.24 units less than the model that included only the `model` and `band` features. These results indicate not only that the strength of correlation between the predicted and recorded responses varies based on the language that the participant is listening to, but also that difference in predicted-recorded correlation between English and Catalan excerpts also varies with model type. This is shown most clearly in Fig. 8a: the average predicted-recorded correlation for models containing

**Subject SD010, Channel LHH5, δ Response**

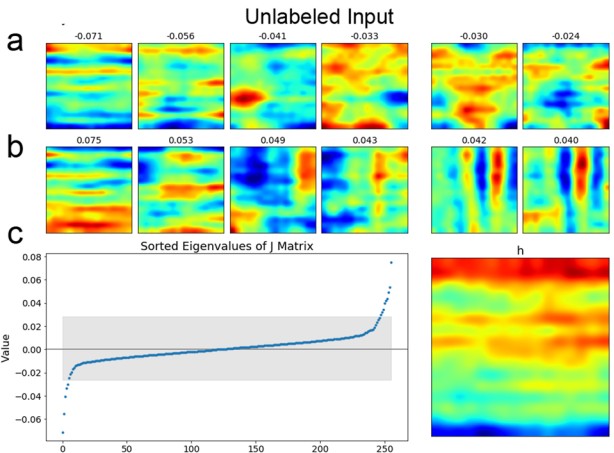

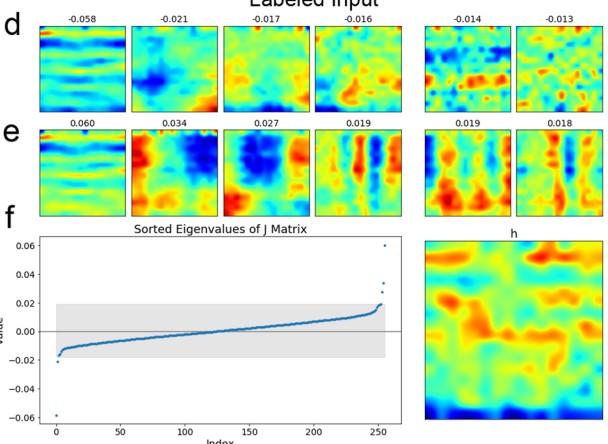

**Fig. 6 | Reconstructed receptive fields reflect available phonemic and spectrographic information. a** Receptive fields corresponding to the six most negative eigenvalues for a single electrode whose delta power response was fit with purely spectrographic information. **b** Receptive fields of the six most positive eigenvalues for the same electrode as (**a**). **c** Left: Eigenvalues of the fit model, ordered by their value. Gray shaded box encompasses the bottom 95% of eigenvalue magnitudes. Right: The power-dependent average receptive field. **d** Receptive fields for the six

most negative eigenvalues reconstructed from the delta power response of the same electrode shown in (**a**) using a model that was fit with both spectrographic and phonemic label information. **e** Receptive fields of the six most positive eigenvalues for the same electrode as (**d**). **f** Left: Eigenvalues of the fit model, ordered by their value. Gray shaded box encompasses the bottom 95% of eigenvalue magnitudes. Right: The power-dependent average receptive field.

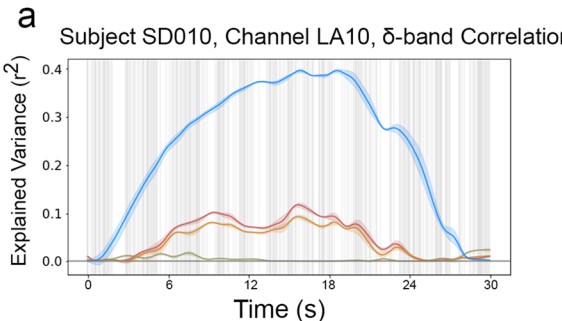

"…says you have to do this in order to get up there they don't take into consideration the attributes the person has so now we're sayin okay if you bring this attribute to the table eh you can help the team out then why not reward you instead of putchin you over here in say a managerial position which you really don't have any call for but you're there because of the money only kay so we're trying to fit the attribute to the person and compensate them accordingly which hey I think we did that we'd prolly…"

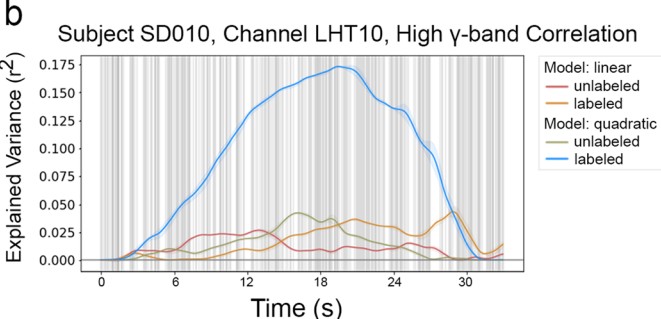

"…think later on when he gets a little older he will be y'know hey cool y'know when he can wrestle with 'im and all that right now he's just he sits there and when he cries he goes whaddas he want like I know go down the list hun change his diaper gi-- if he-- it's been three hours since he's had a bottle g'im a bottle y'know play withim take him in on the kitchen floor so he can roll around cuz our kitchen is like open and there's like nothing in it an uh y'know like go down the list that's what I do he just thinks I automatically know…"

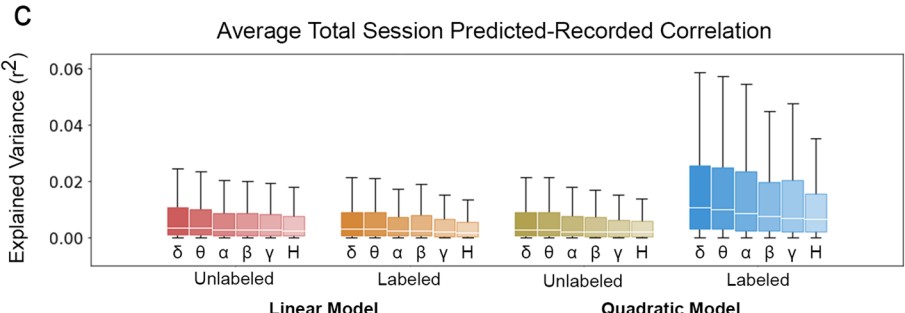

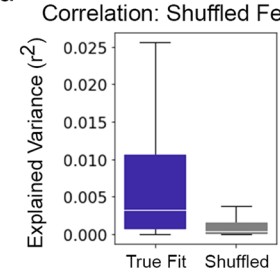

**Fig. 7 | Combined with phonemic label information, stimulus covariance structure best predicts the neural response to speech across all response bands.** Time-varying variance explained by the MNE models for the delta (**a**) and high-gamma (**b**) power responses of single electrodes, illustrating periods of sustained advantage of the labeled quadratic model (blue) over other models. Gray vertical lines indicate phoneme boundaries for the transcripts provided

below each plot. **c** Average variance explained by the MNE models for all subjects, broken out by model type and neural response band ($n$=1255 electrodes per box-plot). **d** Average variance explained by the MNE models for all true fit models vs. all shuffled models ($n$=30,120 electrodes per boxplot). In (**c**) and (**d**), white horizontal lines indicate distribution medians, colored boxes show the IQR, and black whiskers extend a further 1.5 IQR.

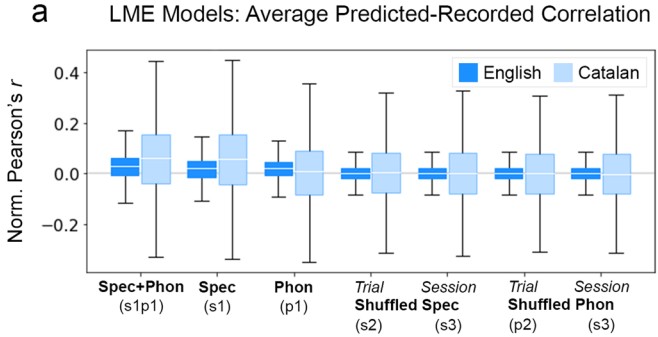

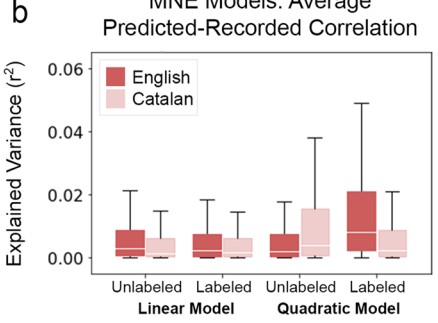

**Fig. 8 | Phonemic labels improve model fits for English but not Catalan data.**
**a** Average correlation between the response predicted by each of the LME models and the actual recorded response; *n*=32,317 for each English boxplot, and *n*=12,286 for each Catalan boxplot. **b** Average correlation between the response predicted by each of the MNE models and the actual recorded response; *n*=7530 electrodes per boxplot. In both (**a**) and (**b**), white horizontal lines indicate distribution medians, colored boxes show the IQR, and black whiskers extend a further 1.5 IQR.

spectrographic features (s1p1 and s1) is higher for Catalan excerpts than for English excerpts. However, for the model containing only phonemic label features (p1), the average predicted-recorded correlation is higher for English excerpts than for Catalan excerpts. Specific language knowledge is required for phonemic label information to substantially predict neural response to speech.

Similarly, to assess the role of specific language knowledge on the fit of the MNE models, a family of models was fit. These models predicted the Fisher Z-transformed Pearson correlation between predicted and observed neural responses on the basis of model order (linear or quadratic), the availability of phonemic label information (yes or no), excerpt language (English or Catalan), and combinations of their interactions. Subject and channel were included as random effects. The model that included `band`, `model`, `label`, and `model:label` features as fixed effects was treated as the baseline model. The formulas for the six models in this family are given in Supplementary Table 3.

As before, model selection was determined using the AIC. Of the six models, the model containing the simplex features `model`, `label`, `lang` and all their interactions best fit the data, having an AIC score 775.87 units lower than the next best model and carrying 100% of the cumulative model weight. Among the models with lower fit, those including a feature for the interaction between excerpt language and label status (`lang:label`) better fit the data than models without this feature, and all models including the excerpt language feature (`lang`) better fit the data than the baseline model. These results indicate that participants' knowledge of the language they are listening to significantly impacts model fit and does so in a way that interacts primarily with the availability of categorical phonemic information. As shown in Fig. 8b, the average predicted-recorded correlation for quadratic models is higher for Catalan excerpts than for English excerpts when the MNE model's input features are purely spectrographic, whereas for MNE models that include phonemic label information in addition to spectrographic information, the average predicted-recorded correlation is higher for English excerpts. In this way, the availability of phonemic information does not benefit MNE model fit for neural activity recorded while participants are listening to an unfamiliar language that they cannot understand. Categorical speech sound information requires specific language knowledge in order to account for neural activity.

## Discussion
In this study more sites sensitive to phonemic identity were observed than would be predicted by chance. While simple in its articulation, this result has far-reaching consequences for phonological theory and theories of language representation in the brain.

These findings support a reallocation of probability mass away from a number of common ideas about the nature of phonology and phonological processing. For example, implicit in many theories of speech production and processing is the assumption that language-specific grammar only occurs at the level of syntax. Phonological knowledge, including knowledge of language-specific phonotactics, is cast as epiphenomenal of lexical knowledge: speech perception is more or less a word recognition problem, and speech production proceeds in a universal and deterministic manner from articulo-acoustic content specified in the lexicon. This study provides evidence that these common assumptions are inaccurate. Language-specific phonological grammar guides the neural response to speech, and it does so at a level of granularity commensurate with the phoneme.

To see how this conclusion follows from the observed data, consider what one would predict if phonemes were not a relevant unit of organization for language in the brain. To maintain true skepticism about the cognitive reality of phonemes, one would need to envision a state of affairs in which underlying representations of speech sounds were altogether unnecessary. That is, at no point in speech processing would a listener assign the value /d/ to some taps and /t/ to others. However, if this were the case and taps were not differentiated in underlying representations, it would be unexpected to observe any phonological underlying sites given the comparisons used in this paper. Instead, one would expect to observe acoustic surface sites only.

We argue that the most parsimonious explanation for the observation of phonological underlying sites is language specific phonological knowledge of the kind generally assumed by working phonologists. More precisely, the existence of phonemic underlying sites suggests that the representations of speech sounds and words are not merely amalgamations of pronunciations that a listener has encountered before. Rather, there is a degree of abstraction between surface, acoustic forms and the prelexical forms that speech sounds are mapped onto in the brain. The nature of this abstraction is language specific, reflecting not only differences in the phonemic inventories of different languages, but also the language specific, contextually-sensitive sound alternations that comprise a specific language's phonological grammar.

The surrounding phonological context is essential to this abstraction. For example, as prefaced in the Introduction, when either /d/ or /t/ occurs following a stressed syllable and between two vowels, their acoustic contrast is neutralized, and both are pronounced as a coronal tap. The context of the /d/ or /t/ is critical to this alternation. Without a preceding stressed vowel and following unstressed vowel, this alternation does not take place. In this way, the surrounding phonological environment provides cues to the underlying

phonological identity of sounds. Although in the case of coronal stop neutralization, the cues may or may not uniquely disambiguate the phonemic identity of the /d/ or /t/ (see Supplementary Fig. 1), in general, phonological abstraction requires that a listener be able to treat the same acoustic information (e.g., a tap) differently based on its surrounding phonological context. We argue that this very process constitutes phoneme identity extraction. For this reason, the point of the baselining in Section 2 was to ensure that what is observed in the time window of interest is not merely the sum of what came before it. While surrounding context undoubtedly plays a role in phoneme identity extraction, one of the key things that this paper shows is that some responses to acoustically similar sounds (i.e., various taps) diverge in a way that is consistent with the extraction of phonemic information and not with pure acoustics.

Dresher[29] lays out three ways that phonologists have conceptualized of the phoneme as an object of inquiry. One class of definitions characterizes the phoneme as a *physical reality*. Representative definitions of this genre describe the phoneme as a language-specific "family" of sounds that "count for practical purposes as if they were one and the same"[30] (via Dresher[29]), perhaps because "the speaker has been trained to make sound-producing movements in such a way that the phoneme features will be present in the sound waves, and [the speaker] has been trained to respond only to these features"[31]. The second class of definitions views the phoneme as a *psychological concept*. This class of explanation is often presented as an alternative to the physical reality of the phoneme: if a physical constant coextensive with the phoneme does not exist, then perhaps a mental constant does instead. Finally, if both the physical and psychological reality of the phoneme are rejected, then Dresher,[29] echoing Twaddell,[32] concludes that the phoneme is a convenient *theoretical fiction*, without material basis in the mind, mouth, or middle ear.

This study addresses the second of these three positions, investigating whether the commonly held understanding of the phoneme as a unit of language-specific contrast has psychological reality or merely theoretical utility. In demonstrating the existence of sites sensitive to phonemic contrast in the absence of acoustic distinction, this study provides support for the classical understanding of the phoneme as a psychological entity and a core level of sublexical linguistic processing. At the same time, it suggests that some theoretical implementations of the phoneme as a unit characterized by minimal (as opposed to exhaustively-specified) sets of distinctive features may have more theoretical utility than psychological reality (see Supplementary Fig. 8). To the extent that linguistic and psychological theories of phonology intend to account for the same sets of behavioral phenomena, these results nevertheless support the continued use of phonemic units in phonological theory and reify the existence of language-specific sublexical structures in language processing.

The morphological results of this study also engage with foundational principles of sublexical language processing. In observing more surface and morphological underlying sites than would otherwise be expected by chance for both the plural and past tense comparisons, this study provides evidence for two interrelated processes in the neural basis of morphology.

It is generally accepted that morphological decomposition takes place during the processing of regularly inflected forms in English and related languages[33–36;cf.37]. The presence of morphological underlying sites in this study is generally consistent with these results. However, the paradigms typically employed to assess the compositionality of morphological units within words arguably gauge fairly abstract proxies for morphological structure. Many rely on priming and assess compositionality through metrics such as reaction time or expectation violation response. The evidence for morphological abstraction presented here has the advantage being gathered during a naturalistic listening task and straightforwardly comparing the difference in neural response to sounds that bear morphological exponence to those that do not carry any morphological meaning.

From this perspective, the difference in response between non-plural word-final [z] and plural [s] or [z] can be attributed straightforwardly to the morphological exponence of [s] or [z], without appeal to intermediary cognitive phenomena such as priming or expectation. That is, the meaning of 'plural' associated with the sounds [s] and [z] drives the response of the morphological underlying sites for the plural comparison, and likewise, the meaning of 'past' associated with [t] and [d] drives the response of the morphological underlying sites for the past tense comparison. This, in general, is a great strength of the paradigm employed in this study. However, these results themselves do not provide explicit evidence that the structure of regular plural and past tense forms is compositional, since it could be the case that, for instance, plural [s] and [z] pattern together at morphological sites through an analogical process. In such a case, plural [s] and plural [z] would still evoke similar neural responses because they index a common morphological exponent, but that commonality would be mediated through an analogical process via the lexicon rather than a compositional, syntax-like process within the word itself. Moreover, this study does not address whether the underlying sites that we observe for the plural and past tense are sensitive only to the regular plural and past tense (i.e. exponent specific) or to "plural-ness" more generally. Determining this would require a follow-up study that includes irregular forms. Nevertheless, the results of this study provide evidence of an early neural response to the presence of particular morphological exponents.

Furthermore, these results support the idea that morphological identity is abstracted over phonologically distinct alternants in a structured, language-specific manner. As was argued for the phonemic sites identified by the tap comparison, the structured relationships between speech sounds and their phonological contexts form a language-specific grammar of sounds. Given their interface with meaning, morphophonological alternations in particular have been of central importance to the development of phonological theory (see Kenstowicz and Kisseberth[14]) and are often appealed to in classic texts as foundational evidence for the existence of the phoneme, since it is through the ways that sounds either change or fail to change the meanings of words that phonemic contrast is most strikingly established. In this way, the existence of morphological underlying sites identified by the plural and past tense comparisons demonstrates both that morphological identity is abstracted over distinct, phonologically conditioned alternants and that morphological exponence transcends phonemic particularities.

Language has numerous levels of analysis that interact in a variety of structured ways. This paper has focused on the dissociation of phonetic, phonological, and morphological levels of analysis through the lens of acoustic features, phonemic labels, and morphological exponents. However, more abstract levels of structure are also at play during natural speech listening, including lexical knowledge, multi-word semantic and syntactic structure, and discourse, as well as more general awareness of language's statistical structure. Given that all these levels of structure are present in speech simultaneously, it is reasonable to wonder how confident we can be that what appear to be phoneme-sensitive responses are really phoneme responses and not lexical, semantic, or some other kind of response.

Confidence that the reported results reflect phonological and not lexical or semantic information comes from the analysis design. In particular, LME and MNE models show that phonemic label information accounts for variance in the neural response that is not accounted for by the acoustic feature set. That is, these analytical approaches do not show evidence of a phonemic response per se; rather, they show that phonemic label features explain aspects of the neural response not otherwise explained by acoustic features. Thus, to be confident that these results are properly phonological and not lexical or

semantic, one need only be confident that the phonemic label features and models do not implicitly have access to lexical or semantic information. Addressing the first of these concerns, phonemic labels are generally free of meaning (see the *arbitrariness of the sign*[38]). Addressing the second, LME and MNE models do not use sufficient sequential information to approximate wordlike information. Together, these facts provide confidence that the reported results reflect sublexical, phonological structure and not lexical or supralexical structure.

Determining the nature and origin of the neural processes that support language processing is crucial for building a mechanistic understanding of linguistic cognition. In this study, band-limited power was examined for six frequency ranges. However, the interpretation of this measure, including its relation to other neural measures and its cellular origins, varies depending upon where in the brain it is recorded.

In intracranial recordings, LFP recorded from gray matter is credited primarily to local synaptically-induced current flow. On the other hand, LFP recorded from white matter is understood to originate from a combination of cortical activity spread by volume conduction into adjacent white matter as well as current flows through the voltage-gated channels that mediate the travel of action potentials along axons[39]. Thus, while the dominant components of electrical activity recorded in gray matter result from the activity of a spatially restricted set of cells proximal to the electrode contact, activity recorded from white matter additionally contains signal that originates from a combination of cells, possibly separated by great distance, that happen to project axonal processes within proximity of the electrode contact. Additionally, the volume conduction properties of white and gray matter differ. Whereas LFP in gray matter (passively) propagates more-or-less equally in all directions—that is, both parallel and perpendicular to the cortical surface—propagation in white matter is directionally biased by the higher density of axonal myelin, such that passive spread occurs more readily along myelin sheaths rather than across them[40,41]. These key differences in the properties of gray and white matter impact the functional interpretation of electrophysiological measures collected from these tissues.

Thus, while activity recorded from white matter is certainly physiologically valid, its interpretation differs from activity recorded in cortical tissue. At present, the electrophysiological properties of white matter are relatively under-explored, and consequently, activity recorded in white matter is considered difficult to interpret and likely under-reported when it is recorded. Accordingly, while this paper does not speculate on how significant sites fit into larger research narratives about the language network, it nevertheless reports results that are primarily driven by activity recorded in white matter. In doing so, this paper contributes to an early understanding of what language-related electrophysiological activity looks like when recorded from these sites.

Relatedly, it is a priori unclear whether the time course of language-related activity recorded in white matter ought to follow the same time course as activity recorded from the scalp or gray matter. Some work suggests that the acoustic processing of phonetic detail takes place within 100–200ms following the onset of a speech sound[42,43], activity related to speech sound categorization roughly 300–500ms after the sound's onset[43], and morphological processing occurs even later[44]. However, other work suggests a messier temporal story, where both categorical and gradient speech sound information are represented simultaneously in the neural signal[45,46], along with morphological information[46,47]. To the extent that it is possible to compare results observed in EEG, MEG, and ECoG data to those observed in SEEG contacts embedded in white matter, the results of this study appear to support the messier story, as shown in Supplementary Figs. 9, 10.

For the coronal tap comparison in particular, the majority of sites that show an acoustic pattern occur in the gamma power and high-gamma power bands. In these bands, the time windows where

significant acoustic responses occurred correlate highly with the windows where significant phonemic responses also occurred (Supplementary Fig. 9). These results suggest that both phonemic category and acoustic detail are processed simultaneously and overwhelmingly separately, since relatively few sites across comparisons exhibited both acoustic and phonemic patterns or surface and morphological patterns (Supplementary Figs. 3, 10).

In fitting mixed effects models containing acoustic features and phonemic label features across the five classic oscillatory frequency bands, the analyses described in Section 2 provide a basic intracranial validation of results found in scalp EEG work by Di Liberto et al.[17]. Di Liberto et al.[17] fit multivariate temporal response functions (mTRFs) to bandpassed scalp EEG data using spectrographic features, phonemic label features, and both spectrographic and phonemic label features. As in this study, they found that models containing both spectrographic and phonemic label features provided the best fit model for delta and theta bands. Furthermore, in these lower frequency bands, their phonemic label model outperformed their spectrographic model. Conversely, in higher bands (alpha, beta, gamma) the model containing only spectrographic features outperformed the model containing only phonemic label features. Validation of these basic results from Di Liberto et al.[17] thus provides useful validation of the dataset used in this study, in light of the novelty of other analyses described in Section 2.

However, the LME analyses in Section 2 also build upon the findings of Di Liberto et al.[17]. In using the AIC to inform model selection, this study accounts for differences in model fit which in Di Liberto et al.[17] could be attributed to differences in numbers of model parameters. This analytical choice provides greater confidence that the observed best-fit models fit best due to the substance of their features rather than their dimensionality. Additionally, in an alternative to the LME analysis presented in Section 2, the spectrographic features used to fit LME models were drawn from the latent space of a variational autoencoder, as described in Supplementary Fig. 11. In this sense, like the phonemic label features, the spectrographic features in these models were also abstracted from surface acoustics. However, whereas the phonemic label features represent a linguistically informed compression of the acoustic signal, the compressed spectrographic features are purely informed by acoustic information. It is remarkable then, with all their compression, that the weightings of these feature sets are still capable of reconstructing neural response with such high fidelity, with a correlation of over 0.4 at times. In this way, the ability of these features to account for a significant portion of neural variance validates the use of such nonlinear compression methods for neural encoding analyses.

Moreover, in addition to validating Di Liberto et al.'s[17] main results, this study validated one of the finer details of their results. In particular, Di Liberto et al.[17] found that the overall correlation between predicted and recorded responses was strongest in the lowest frequency bands and dropped to a fraction of that magnitude as center band frequency increased. This effect can be seen clearly in Fig. 5e and to a lesser extent for the MNE models in Fig. 7c. Di Liberto et al.[17] attributed this steep dropoff in the predictive ability of their models to the lower signal-to-noise ratio of these bands when recorded from the scalp. Observing this effect in intracranial recordings is not necessarily inconsistent with Di Liberto et al.'s[17] interpretation, but the sources of noise may be different. For example, it may be the case that higher frequency signals have lower fidelity in white matter sites due to the distance of these sites from signal generators.

The variability in the best-fit LME model across participants for gamma power and high-gamma power data is not easily explainable. Properties of the higher bands themselves may be in part responsible for this variability. Because higher frequencies attenuate more quickly over distance than lower frequencies, high-frequency components of the signal at each electrode reflect more local responses while lower-

frequency components are integrated over larger anatomical space. Thus, all else being equal, heterogeneity across sites whether within or between participants may more easily impact LME fits in higher frequency bands than in lower frequency bands, since the latter are likely capturing a more global (i.e., shared) response. This may contribute to the fact that there is more participant variability for higher bands than for lower bands.

Variability within particular bands may result from differential electrode coverage across participants. The fact that both SD018's gamma power and high-gamma power were best fit by the s1p1 model would be consistent with such an explanation, since electrode coverage is a factor that is fixed for each participant, and we might therefore expect coverage-based differences to be consistent within participants. However, the fact that only SD013's gamma power and only SD011's high-gamma power are best fit by the s1p1 model reveals the true complexity of a coverage-based explanation for the variability in these results. For the best fit model for SD013's gamma power to diverge from that of all other participants for only that power band would require coverage that differs from other participants' in such a way that it differentially impacts gamma power only. Similarly, the fact that the best fit models for SD018 differ from the majority in only two bands requires one to posit a state of affairs in which SD018's coverage differs in a way that substantially impacts only two bands.

Individual variation presents another possible explanation. Whereas a coverage-based explanation assumes that results across participants ought to be the same and only differ because non-homologous sets of neurons were recorded across participants, an explanation based on individual variation would allow for the possibility that even with functionally identical coverage, individuals may still vary in the way that they process speech sounds. From this perspective, the consistency across participants in lower bands is merely artifactual of the small number of participants whose data are reported here, and perhaps if data from a greater numbers of individuals were available, some variability in the best-fit model would be observed in each band. However, while individual variability most certainly exists in speech sound processing, it seems unlikely that information about speech sound categories and their acoustics, being both quite close to the sensory periphery and fundamental to speech perception, would vary substantially across individuals. This is an empirical question, though, and while its answer ought to be pursued, it is not a question that this dataset is well-suited to address. Given the limitations of the dataset, arriving at a satisfactory account for the cross-participant variability of these results will require follow-up work.

Finding that phonemic label features did not improve model fits for Catalan speech provides further support for the claim that phonemic label features do not simply reinforce information available in speech acoustics. Rather, phonemic label features account for neural activity that requires specific language knowledge.

The poor performance of phonemic label features for Catalan speech is expected despite the overall similarity in sound inventories between Catalan and English. This is because patterns of alternations in Catalan differ from those in English, such that listeners unfamiliar with Catalan phonology cannot be expected to extract accurate Catalan categories without knowledge of the patterns. For example, English speakers may interpret Catalan alveolar taps as either /d/ or /t/, due to the English coronal neutralization pattern discussed in Section 2; however, Catalan has no such pattern, and its taps are phonologically related to neither /d/ nor /t/. For this reason, Catalan phonemic labels were correctly predicted to poorly explain the neural response to speech for English listeners listening to Catalan.

However, it was unexpected that the success of the acoustic features explaining the neural response to Catalan speech exceeded that of English speech. Two interrelated factors could explain this unexpected result. First, participants were asked to listen carefully for a recognizable word in the Catalan speech, but were not directed to listen for any particular word in the English speech. For this reason, increased attention to the speech stream could account for the overall higher performance of the acoustic feature sets in explaining the neural response to Catalan speech. Relatedly, as an unfamiliar language, the faithful tracking of speech acoustics may occupy more of the neural response to Catalan speech than to English, a language for which all participants have much more robust predictive models.

MNE models model the relationship between the stimulus and neural response as a linear combination of the contributions of stimulus feature sets with varying cardinalities. That is, a first-order MNE model models neural response as the contributions of individual stimulus features; a second-order model sums single feature contributions with the contributions made by pairs of stimulus features; a third-order model further includes the contributions of triplets of stimulus features; and so on.

This study specifically compares the fits of first- and second-order models. In doing so, it assesses the extent to which relationships between stimulus features contribute to model goodness-of-fit beyond the contributions of individual features themselves. In finding that second-order, quadratic models account for significantly more of the variance in neural response only when phonemic label information is available, this study shows that categorical phonemic information bootstraps acoustic information in explaining the neural response to speech. More precisely, without the availability of pairwise information about the stimulus, phonemic label information does not contribute significantly to predicting neural response.

This result provides structure to long-held understandings of the important role that relative spectrotemporal features play in speech sound identification. For example, at single time points, measures of F1:F2 ratio or spectral tilt provide key information for discriminating among vowels or among fricatives, respectively, and comparing measures across time provides further discriminatory ability. Features of spectrographic stimulus covariance have also been implicated in speaker normalization[48]. In these ways, the sensory signal itself provides an exceptional degree of structure that mirrors in fine scale what linguistic categories demonstrate at a coarser scale. That is, speech sounds are built from relationships between acoustic features; phonemes from relationships between speech sounds; morphemes from relationships between phonemes; meaning from relationships between morphemes, and so on.

This study provides evidence that relative power between pairs of time-frequency bins supports the neural response to speech when phonemic category information is also available. Linking features of the sensory signal to linguistic categories in this way provides structure for an early link between sensory and linguistic cognition.

## Methods

### Participants

Study participants were ten patients at UC San Diego Health who underwent intracranial stereo EEG and subdural electrode implantation as part of treatment for refractory epilepsy or related conditions. All were native English speakers, who had no prior experience with Catalan, and all reported normal hearing and performed within the acceptable range on a battery of neuropsychological language tests. The research protocol was approved by the UC San Diego Institutional Review Board, and all subjects gave written informed consent prior to surgery. Basic patient information is summarized in Table 2.

### Participant gender

Four of the ten participants were women, and six were men. Participant gender information was assigned impressionistically by researchers and recorded only for demographic purposes. Gender was not considered during study design and did not factor in participant recruitment.

**Table 2 | Summary of basic patient information**

| Patient | Age | Gender | Hand. | Wada | Coverage | Language Experience |
|---------|-----|--------|-------|------|----------|---------------------|
| SD010 | 20s | M | R | – | sEEG: LH,RH | English |
| SD011 | 30s | M | R | LH | sEEG: LH,RH | English, Creole, French |
| SD012 | 20s | F | R | LH | sEEG: LH,RH; Grid,strips: LH | English |
| SD013 | 40s | M | L | LH* | sEEG: LH,RH | English, Spanish |
| SD015 | 50s | F | | – | sEEG: LH,RH | English |
| SD017 | 20s | M | R | – | sEEG: RH | English, Spanish |
| SD018 | 40s | F | L | RH | sEEG: LH,RH | English, Spanish |
| SD019 | 20s | M | R | – | sEEG: LH,RH | English |
| SD021 | 30s | F | R | – | sEEG: LH,RH | English |
| SD022 | 20s | M | R | – | sEEG,Grid: RH | English |

*While patient SD013 did not experience a clear, prolonged speech arrest with either injection, he demonstrated greater language deficits following left hemisphere injection.

## English Stimuli

Participants passively listened to short excerpts of conversational American English speech taken from the Buckeye Corpus[49]. To assess participant attention to the task, participants responded orally to a two-alternative question about the content of each English passage after they listened to it.

Passages were 25-76s long (mean 38s) taken from 27 (12 women; 15 men) native speakers of Midwestern American English living in Columbus, OH between October 1999 and May 2000. Per corpus documentation by Pitt et al.,[49] excerpts were recorded monophonically on a head-mounted microphone (Crown CM-311A) and fed to a DAT recorder (Tascam DA-30 MKII) at a 48kHz sampling rate through an amplifier (Yamaha MV 802). Following each excerpt, participants heard a two-alternative choice question regarding the content of the passage they heard, and they were asked to respond orally to the question to ensure that they were alert and paying attention to the passages that they heard. These questions were recorded with a Blue Yeti USB microphone sampling at a rate of 48kHz by a speaker of American English. The amplitude of all English passages and questions was normalized to -20dBFS prior to padding with 500ms of silence.

Transcription, segmentation, and labeling were performed for all acoustic stimuli. Both orthographic and phonetic transcriptions were created to provide human-readable text of the speech stimuli. Segmentation created time codes for boundaries between units, and labeling created labels for the spans between boundaries. All stimuli were segmented and labeled for word, part of speech, and phone identity. In this way, labels for words, parts of speech, and phones were time-aligned to the acoustic stimuli. Transcription, segmentation, and labeling procedures for passages from the Buckeye Corpus are described in Pitt et al.,[49] and transcription, segmentation, and labeling of task instructions and content questions was performed by a phonetically-trained researcher at UC San Diego, using the protocols detailed in Pitt et al.[49].

## Catalan stimuli

Interspersed with the passages from the Buckeye Corpus were short passages of conversational Catalan taken from the Corpus del Catalá Contemporani de la Universitat de Barcelona[50]. To assess participant attention to the task, participants were instructed to listen for English words embedded in the recording and press the space bar when they heard an English word.

Catalan passages were 44-55s long (mean 49s) taken from 2 (1 woman; 1 man) native speakers of Catalan—one woman from Benabarre, Huesca and one man from Tamarit, Catalonia. Three passages were excerpted from each speaker for a total of six Catalan excerpts, and Catalan excerpts were interspersed into the passive listening task such that one Catalan excerpt was included in each block of the task. Since patients are not expected to understand Catalan speech, two-alternative questions about the content of the Catalan passages were not suitable to assess participant attention. Instead, English nouns were digitally spliced into the Catalan excerpts, and participants were asked to press the space bar when they heard an English word. For each Catalan passage, three English nouns were excised from the Buckeye corpus digitally spliced into the Catalan recording. These English nouns were taken from one female speaker and one male speaker from portions of the Buckeye Corpus not otherwise heard in the passive listening task. Nouns were excised from their original contexts and inserted into the Catalan speech at zero crossings to minimize acoustic artifacts. Inserted nouns were gender-matched with their Catalan carrier passages, but no further controls were taken to match for voice similarity. Prior to insertion into Catalan speech, the English nouns were normalized to -20dBFS, and after insertion into Catalan speech, the entire mixed-language passage was normalized to -20dBFS. This process of amplitude normalization resulted in English nouns which were clearly audible and segregable from the Catalan speech stream. Nouns were inserted into the Catalan speech during natural pauses at roughly equal but not regularly spaced intervals to increase uncertainty and encourage attention to the unintelligible speech stream.

Catalan stimuli were orthographically and phonetically transcribed, segmented, and labeled. Phonetic and orthographic transcriptions for Catalan passages were archived alongside audio recordings in the Dipósit Digital de la Universitat de Barcelona (http://diposit.ub.edu/dspace/handle/ 2445/10413), and the passages used in this study were subsequently segmented and labeled in Praat[51] by a phonetically-trained researcher at UC San Diego using the phonetic transcriptions provided with the corpus. Given the similarity in the phonetic inventories of Catalan and English, the segmentation criteria used for the Buckeye corpus (reported in Pitt et al.[49]) were also used to segment the Catalan passages used in this study.

## Data acquisition and processing

Experimental instructions and stimuli were presented to participants in their hospital rooms on a Windows 10 desktop PC (Dell XPS 8910) using PsychoPy for Python 2.7[52,53]. The task was conducted in six blocks, each of which contained eight English trials and one Catalan trial. The average block duration was 6 minutes 30 seconds. All participants completed at least two of the six blocks, and the average participant completed four blocks, listening to 26:45 min of speech totaling 15,070 phones. Each trial consisted of a short, spoken passage followed by a content question and an oral response.

Intracranial EEG signals were amplified using a multi-channel amplifier system (Natus Quantum) and recorded using Natus NeuroWorks software. Auditory stimuli and oral responses were recorded simultaneously with the EEG data by feeding the output of a Zoom H2n microphone as an additional input channel to the Natus Quantum amplifier.

After recording, neural data were deidentified and exported from the clinical NeuroWorks system in .edf (European Data Format) format

for pre-processing using the Python package MNE Python[54]. Channels displaying excessive artifacts or line noise were removed (45 channels total). Remaining channels were common average referenced, notch filtered at 60Hz and its harmonics, bandpass filtered 0.1-170Hz, and downsampled to three times the lowpass cutoff (510Hz). Independent Component Analysis (ICA) was used to remove stationary artifacts from the filtered data, and time intervals containing remaining artifacts were visually identified and discarded. Channels selected for analysis were those which exhibited reliable evoked response to speech stimuli, determined by a sliding window t-test between responses to randomly-selected time frames during the passive listening task and in silence ($p < 0.05$).

Open source Python (3.6.5) and R (3.6.3) libraries were used to analyze the data in this study. Standard functions in MNE Python (0.20.8) were used for data preprocessing. LME models were constructed and fit using the R library `lme4` (1.1-21). MNE models were constructed and fit using the Python package `pyMNE` (https://github.com/MarvinT/pyMNE).

## Band power
The frequency bands used in this study were delta (1–3Hz), theta (4–7Hz), alpha (8–12Hz), beta (13–30Hz), gamma (31–50Hz), and high-gamma (70–150Hz) bands. To compute the power for each band, the analytic amplitude from eight Gaussian band-pass filters with logarithmically increasing center frequency[55,56] was averaged. For the ERP analyses, targets were defined as phones involved in one of the three language-specific comparisons, and band power data were then segmented into peri-target epochs with 100ms pre-target and 500ms post-target, and each epoch was z-scored relative to the mean and standard deviation of its 100ms pre-target baseline.

## Electrode localization
Stereo EEG electrodes were localized by registering each patient's pre-Op T1-weighted MR volume to an interaoperative CT in 3dSlicer[57,58] and manually marking the contact CT artifacts.

## Significant electrodes
Broadband neural activity was recorded from a total of 1355 valid channels across the ten subjects, and for each functional band, each channel was assessed for speech responsiveness using a sliding-window one-way t-test, where the band power for 500ms silent epochs was compared against that of an equivalent number of randomly sampled speech epochs. Valid channels were those that were not discarded due to the presence of excessive line noise or artifacts. Baselining was performed by subtracting from each epoch the average channel response for the 100ms preceding the epoch onset, and t-tests were calculated for nine 100ms windows with 50ms overlap across the 500ms post-baseline portion of the epoch. Within each window, values for each token were averaged over time. Channels were considered speech responsive if the t-test for at least one window was significant with $p < 0.05$.

## Spectrographic features for LME models
For each timepoint of the response, a set of features representing the preceding acoustic context was calculated. First, each excerpt was resampled to 16 kHz, and a spectrogram was computed for the downsampled file using the `spectrogram` function from the `scipy.signal` Python library. For each excerpt, this function calculated consecutive Fourier transforms over segments of the excerpt waveform that were 128 time bins (=8ms) in length with no zero padding. Segments were windowed using a Hann function and had 64 time bins of overlap with one another. This function resulted in a spectrogram with 65 frequency bins. The DC component of the spectrogram was removed, and the remaining 64 frequency bins were log transformed and then pairwise-averaged twice, resulting in a spectrogram with 16

frequency bins. Time bins were then pairwise-averaged three times, such that each time bin of the final spectrogram represents a portion of time eight times longer than the initial resolution (=64ms). Finally, the full spectrogram, representing the entire excerpt, was split into overlapping 16 × 16 chunks, such that each time bin of the total excerpt spectrogram—from the sixteenth bin to the final bin—could be mapped to a 16 × 16 (=1024 ms) chunk representing the spectrographic content immediately preceding that time bin.

The final sample rate of the spectrogram time bins was 15.625Hz. However, all preprocessed neural data had sample rates of 510Hz, meaning that each spectrogram time bin corresponded to between 32 and 33 timepoints of the neural data. Therefore, to place the spectrographic features and neural response in one-to-one correspondence with one another, for each spectrographic time bin, the samples of the neural response that took place during that time bin were averaged together.

As 16 × 16 chunks, each spectrogram would contribute 256 features to the mixed effects model, a computationally prohibitive number of features to fit over hundreds of thousands of datapoints. Thus, to further reduce the dimensionality of the spectrographic features without resorting to further averaging and loss of spectro-temporal resolution, two different approaches were taken. Reported in the main body of the manuscript, dimensionality was reduced by using only the 16 × 8 (=128) most recent features, representing the 512ms preceding each time bin. Reported in Supplementary Information, 128-dimensional representations of each 16 × 16 spectrogram were drawn from the latent space of a Generative Adversarial Interpolative Autoencoder (GAIA)[59] that had been trained on these spectrograms.

GAIA networks are a kind of neural network whose architecture is based on a combination of Autoencoder (AE) and Generative Adversarial Network (GAN) architectures. The GAIA model used to generate spectrographic features for the LME models used in this paper was built in Tensorflow 2.2 as a class of `tensorflow.keras.Model`. The AE supporting the generator role of the GAN contained two basic sub-networks, an encoder and a decoder.

The encoder contained an input layer for batches of 16 × 16 tensors followed by two convolutional layers with 32 and 64 filters, respectively. Both convolutional layers had RELU activation, a kernel size of 3, and a 2 × 2 stride. The output of the second convolutional layer was flattened into a single dimensional tensor using a `tf.keras.layers.Flatten` layer and its output was sent to a densely-connected layer of 128 units.

The decoding subnetwork was symmetrical to the encoder, taking the output of the 128-dimensional z layer as the input to a densely-connected layer. The densely-connected layer fed into a `tf.keras.layers.Reshape` layer that returned its input into a multi-dimensional tensor. This layer was then followed by three deconvolutional layers. The first two deconvolutional layers had 64 and 32 filters, respectively, and each used a RELU activation function, had a kernel size of 3, and a 2 × 2 stride. The final deconvolutional layer had one filter, a kernel size of 3, a 1 × 1 stride, and used a sigmoid activation function.

Like the generator, the discriminator network was an AE, but with a different structure. The discriminator AE was a UNET, an end-to-end fully-convolutional AE architecture originally developed by Ronneberger et al.[60] for biomedical image segmentation.

The encoder for this architecture contains an input layer followed by blocks of convolutional and pooling layers. The first block contained two convolutional layers with RELU activation functions, 3 × 3 strides, and 32 filters each, followed by a pooling layer where each unit of the layer downsampled its input by taking the maximum value of its 2 × 2 window over the input. This block was followed by a block containing convolutional layers with 64 filters each but that was otherwise identical to the first block. The two blocks were followed by a final convolutional layer with 128 filters.

Like the generator, the discriminator UNET's decoding network was symmetrical to its encoding network. The output of the final convolutional layer of the encoder was the input to another 128-filter convolutional layer. The output of that layer was fed into the first of two blocks of upsampling and convolutional layers. This block began with an upsampling layer that, for each value of the input, output a $2 \times 2$ tensor of that value. The upsampling layer was followed by two convolutional layers with 64 filters each. The convolutional layers of the second block had 32 filters each, but the second block was otherwise identical to the first block. The final layer of the decoder was a convolutional layer with one filter, a kernel size of 3, a $1 \times 1$ stride, and used a sigmoid activation function.

The Adam optimizer (`tf.keras.optimizers.Adam`) was used to minimize the pixel-wise error for the generator network. This optimization algorithm is a method of stochastic gradient descent that adapts the learning rate for each parameter of the model based on the first- and second-order moments (i.e., mean and variance) of the gradients. The Adam optimizer for the generator network was initialized with a learning rate of 0.001 and an exponential decay rate of 0.5 for the first moment estimates, and all other parameters for the optimizer were set to their defaults.

The optimizer used for the discriminator network was a Root Mean Square Propagation (RMSProp) optimizer (`tf.keras.optimizers.RMSprop`), a stochastic gradient descent algorithm that, like Adam, maintains adaptive learning rates for each model parameter, but adapts learning rates based only on the mean of recent gradient magnitudes. It was initialized with a learning rate of 0.001 as well, and all other optimizer parameters were set to their defaults.

The overall logic of the GAIA architecture is as follows: a standard autoencoder is trained to reproduce its inputs after passing them through a low-dimensional $z$ layer. The 128-dimensional space of the $z$ layer is sampled linearly, sampling two points in the latent space as well as evenly spaced samples—interpolations—between those two points. Both the interpolations drawn from $z$ and the low-dimensional representations of the original inputs are passed through the decoder network, yielding sets of $16 \times 16$ spectrograms—$\tilde{x}_i$ and $\tilde{x}$, respectively. These reconstructed spectrograms ($\tilde{x}_i$ and $\tilde{x}$) are then passed through the UNET autoencoder alongside original spectrograms ($x$), and the network is trained to discriminate between the three types of UNET reconstructions—reconstructions of original inputs ($D(x)$), reconstructions of the first AE's reconstructions ($D(\tilde{x})$), and reconstructions of the interpolations drawn from $z$ that do not correspond to any of the original inputs ($D(\tilde{x}_i)$).

Prior to training, each spectrogram was normalized such that its minimum value was zero, and its maximum value was one. In total, across all excerpts, the dataset contained 68,271 spectrograms, which were randomly split into training and testing sets that contained 85% and 15% of the total dataset, respectively. The GAIA model was trained on batches of 4096 spectrograms, with 14 batches per training epoch, and early stopping was implemented such that training halted when the smoothed generator loss of an epoch exceeded that of the tenth-most recent epoch. To calculate this smoothed generator loss, lists of each epoch's loss values were saved during training to aid in determining when to stop training. To prevent local perturbations in the generator loss from causing training to stop inappropriately early, these lists were smoothed using a Hann function with a window length of 30.

The model trained for a total of 83 epochs. After training, all spectrograms were run through the encoder portion of the generator network, and each spectrogram's 128-dimensional latent space representation was saved for use in the mixed effects model. In all models, this feature set is referred to as the s1 feature set because it contains spectrographic features.

## Phonemic label features

During preprocessing, neural data were temporally registered to stimuli metadata. For each phone the participant heard, the start time, stop time, and duration of the phone were registered, alongside the phone's identity, the identity of the word it was part of, the word's part of speech, and the excerpt it was part of. From this information, a timeseries of one-hot coded features was generated for the set of unique phone labels. In other words, each timepoint of the neural response was assigned a vector of features, where each feature represented a phone label; the value of the feature corresponding to the current phone label was set to 1, and all other feature values were set to 0. Following this feature assignment, the matrix of phone label features was temporally averaged in the same way that neural data were temporally averaged to attain a one-to-one mapping with the spectrographic features. Because each vector of phone label features was sparse and binary, the resultant values in each vector summed to 1. In this way, the values in each feature vector corresponded to the proportion of time within that window where a particular phone was played for the participant. In all models, this feature set is referred to as the p1 feature set because it contains phone label features.

## Linear mixed effects models

For each participant, families of LME models were fit for seven different neural response types, each derived from preprocessed raw recorded data. Six of these response types were the z-scored power for different EEG frequency bands, and the seventh was broadband LFP. The frequency bands used in these models were delta (1–4Hz), theta (4–7Hz), alpha (8–12Hz), beta (13–30Hz), gamma (31–50Hz), and high-gamma (70–150Hz) bands; broadband LFP contained frequencies 0.1–170Hz.

For each of these response types, seven LME models were fit—three models of interest and four shuffle conditions. The three models of interest contained either only spectrographic features (s1), only phonemic label features (p1), or both spectrographic and phonemic features (s1p1). For both the s1 and p1 feature sets, two forms of shuffled feature sets were also created. Feature sets s2 and p2 were created by shuffling s1 and p1 features within excerpts, and feature sets s3 and p3 were created by shuffling s1 and p1 features across the entire recording session.

All s-series models contained 128 features, each corresponding to a time-frequency bin of the spectrogram for the analyses presented in the Main Text. For the analyses described in Supplementary Information, each of these 128 features corresponded to a dimension of the latent space constructed from the GAIA network trained on excerpt spectrograms as described in the section on spectrographic features for LME models in Section 4. However, based on the number of excerpts that the participant listened to, the number of phonemic label features in each model differed slightly. For participants who completed at least four blocks of the passive listening task, there were 72 unique labels included in the p-series models. Models for participants who completed fewer blocks had at least 60 phonemic features. Labels that did not occur in the first two task blocks were uncommon phones and extraneous labels such as ɔɪ, nasalized vowels like aū and eĩ, and laughter.

## Model evaluation

Relative goodness of model fit was evaluated using the AIC. The AIC is an estimate of prediction error, calculated for each model as

$$AIC = 2k - 2\ln(\hat{L}) \tag{1}$$

Where $k$ is the number of estimated parameters in the model and $\hat{L}$ is the maximum value of the model's likelihood function.

The AICw of each model $i$ was calculated as $\exp((\text{AIC}_{min} - \text{AIC}_i)/2)$, where $\text{AIC}_{min}$ is the lowest AIC value of the set of models and $\text{AIC}_i$ is the AIC value for model $i$.

## MNE Input Features

Input features were created for each of the two conditions of interest. For the unlabeled spectrogram condition, 256-dimensional (16x16) spectrograms were created for each timebin of the neural response to represent the 1024ms of acoustic context preceding that timebin. These spectrograms were created using the procedure described in the section on spectrographic features for LME models in Section 4. For the labeled spectrogram condition, 8-bit binary codes (i.e., [01010110]) were created for each possible ARPABET label present in the Buckeye Corpus. Then, two labels (totaling 16 bits) were assigned to each spectrogram. For 1024ms windows containing only one phone-level label, the 16-bit label given to that spectrogram was the simple concatenation of the code for that ARPABET label with itself. For spectrograms containing more than one phone-level label, the 16-bit label given to that spectrogram was the simple concatenation of the code for the label that occupied the plurality of that 1024ms window and the runner-up label. Then the mean and standard deviation of each spectrogram was calculated, and the top row of each spectrogram was replaced with a 1x16 vector representing its 16-bit label. For this row vector, "0" entries of the 16-bit label were replaced with the value one standard deviation below the spectrogram's mean, and "1" entries were replaced with the value one standard deviation above the spectrogram's mean. In this way, spectrographic labels were designed to have minimal impact on the prexisting variance and covariance structure of the spectrograms.

## MNE models

MNE models were fit for each participant for each response type for each condition. These models fit a logistic function that models the response type (LFP or band power) as a linear combination of first- and second-order features of the stimulus (labeled or unlabeled spectrograms).

MNE models minimize the mutual information between the response and the stimulus by maximizing the noise entropy of the stimulus. This is accomplished by rewriting the mutual information between the stimulus and the response as the difference between the response entropy and the stimulus noise entropy. Then, imposing a minimal model on the conditional-response probability to ensure that the stimulus noise entropy is capable of being maximized results in a logistic function that models the probability ($P$) of a neural response ($y$) given a stimulus ($\mathbf{s}$) as a linear combination of first- and second-order features of the stimulus.

Thus, a second-order MNE model takes the form

$$P(y|\mathbf{s}) = \frac{1}{1 + e^{-z(\mathbf{s})}}, z(\mathbf{s}) = a + \mathbf{h}^\mathrm{T}\mathbf{s} + \mathbf{s}^\mathrm{T}\mathbf{J}\mathbf{s} \qquad (2)$$

where $a$ corresponds to the mean unit response; $\mathbf{h}$ corresponds to the neural response constrained by the stimulus variance; $\mathbf{J}$ corresponds to the neural response constrained by the stimulus covariance; and $\mathbf{s}$ corresponds to a set of stimulus features. A first-order MNE model contains only the first two terms of $z(\mathbf{s})$.

To prevent overfitting, model parameters were estimated over four jackknives, where input and response data were split into four batches, and for each jackknife, three of the batches were used as training data, and the remaining batch was used for testing. The log loss between the response and the weighted input was minimized using `fmin_cg`, a nonlinear conjugate gradient algorithm from the scientific computing library `scipy.optimize`, and early stopping was implemented such that if the log loss on the testing data did not

decrease for ten consecutive epochs, training was halted. The optimized weights for each of the four jackknives were then averaged, and this set of mean weights was used in all subsequent work.

For each channel, the fit models were used to generate predicted neural responses. First-order predictions were generated using the fit linear features only, with $z(\mathbf{s}) = a + \mathbf{h}^\mathrm{T}\mathbf{s}$, and second-order predictions were fit using the complete model, with $z(\mathbf{s}) = a + \mathbf{h}^\mathrm{T}\mathbf{s} + \mathbf{s}^\mathrm{T}\mathbf{J}\mathbf{s}$. Then for each predicted response, Pearson's $r$ was calculated to assess the degree of correlation between the recorded and predicted responses. These values were then normalized using the Fisher Z-Transformation such that prediction quality could be compared across model types and conditions.

## MNE shuffled models

To ensure that each model fit better than would be expected by chance, model predictions were also generated from the shuffled features of the fit models. For the linear model, the entries of the 256-dimensional vector feature representing the power-responsive average were shuffled, and predictions were generated using the shuffled linear features only, with $z(\mathbf{s}) = a + \mathbf{h}^\mathrm{T}\mathbf{s}$. For the quadratic model, the linear feature was shuffled as described for the linear model. Additionally, the quadratic model's J-matrix was eigen decomposed, its eigenvalues were shuffled, and the J-matrix was then recomposed using the shuffled eigenvalues. Predictions for the shuffled quadratic models were then generated as expected, with $z(\mathbf{s}) = a + \mathbf{h}^\mathrm{T}\mathbf{s} + \mathbf{s}^\mathrm{T}\mathbf{J}\mathbf{s}$. Shuffling in this way leaves the quadratic features (the eigenvectors of the J-matrix) intact, altering only the weight afforded to each feature in the predicted response. This manner of shuffling therefore provides a relatively high estimate of the quality of response that would be expected by chance, since it preserves the structure of the second-order features in their entirety, merely re-weighting their contribution to the predicted response.

## Reporting summary

Further information on research design is available in the Nature Portfolio Reporting Summary linked to this article.

## Data availability

The datasets generated and analyzed during the current study are available in the OpenNeuro repository, registered with this https://doi.org/10.18112/openneuro.ds004703.v1.0.0. Source data are provided with this paper.

## Code availability

Code and source data for all figures are available at https://github.com/acmai/NCOMMS-23-00607.

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

## Acknowledgements

Thanks to Eric Baković, the members of Gentner Lab, and the AMP 2021 and SNL 2021 poster session audiences for questions, comments, and suggestions on this work. Thank you also to Burke Rosen for assistance with electrode localization. This work was supported by NIMH training fellowship T32MH020002 and William Orr Dingwall Dissertation Fellowship to A.M. Remaining errors are ours.

## Author contributions

A.M.: Conceptualization, Methodology, Investigation, Software, Formal Analysis, Visualization, Writing—Original draft preparation; S.R.: Investigation, Data Curation; S.B.H.: Resources, Project Administration; J.S.: Resources, Project Administration, Funding Acquisition; T.G.: Conceptualization, Supervision, Writing—Review & Editing.

## Competing interests

The authors declare no competing interests.
