## [Peer Review File · Nature Communications]

Acoustic and language-specific sources for phonemic abstraction from speechREVIEWER COMMENTS

Reviewer #1 (Remarks to the Author):

Summary

In this work, the authors try to distinguish between the acoustic and phonemic aspects of the neural response to speech, using the allophones of /t/ and /d/. Similarly, they try to distinguish the neural effects of morphological information of plural and past tense by comparing the word-final phonemes /s/ and /z/ for plural and /t/ and /d/ for past tense. They find that both phonemic and morphological information are represented in the neural signal, especially in the lower frequency band powers. The question the authors are addressing is very important

and debated. However, there are major concerns about the methodological approach to answering this problem.

Major comments

1. The ratio of “speech selective” electrodes, as reported in Figure 1, is unreasonably high, especially considering that the coverage is mostly outside the auditory cortex. I think two incorrect assumptions cause this:

(I) The methods section (5.7 Significant Electrodes) is unclear on what component of the neural signal is being t-tested between speech and silence. Is it the raw signal? The power at a specific frequency band? While, for example, comparing the high-gamma power at an electrode site between speech and silence is meaningful, comparing the raw values is not as meaningful.

(II) It is incorrect to apply the term “speech selective” when the selection method only guarantees that the response of the significant electrode sites differs between speech and silence. I suggest using the term “sound responsive” instead.

2. A primary advantage of intracranial EEG over noninvasive EEG or MEG is superior spatial resolution. Having a reasonably wide-spread coverage, I think the authors should have taken advantage of this fact and report the spatial implications of their findings. While I appreciate the section in the discussion dedicated to the differences between grey and white matter (4.3

Localization & Timing), the authors are in a position to address this in their results.

3. Related to #2, when combining the responses of all electrodes to fit an encoding model (for example Figure 5), the high-frequency bands are at a disadvantage since they capture more local dynamics than low-frequency bands. This would make the LME model discard the local dynamics of a frequency band like high-gamma, which might include the phonological and morphological representations the authors are looking for. This also relates to the discussion raised by the authors in section 4.5 (Cross-participant variability in LME model fits).

4. The usage of the GAIA autoencoder adds a confounding complexity to the problem without any significant payoff. First, a neural network autoencoder compressing 1 second of speech into a 128-dimensional vector can be learning complex “underlying” representations that are not simple acoustics, which importantly, invalidates the results of the LME analysis when comparing s1, p1 and s1p1 (Figure 4). Second, if the amount of compression desired is only 50% (256 -> 128), a linear dimensionality reduction method such as PCA could suffice. Alternatively, shorten the temporal window of the spectrogram from 1024 ms to 512 ms.

5. Related to #4, a potentially complex representation of 1 second of speech spanning multiple words (Figure 4, left) is much more powerful/informative compared to a one-hot encoded phoneme vector. This makes the model comparison unfair. The equivalent purely phonemic model should have the phoneme labels for the last 1 second to be comparable. Since the dimensionality of such a model would be high, perhaps a similar autoencoder network could compress the phoneme labels of the 1-second window into a 128-dimensional vector.

Alternatively, the authors can consider at least giving the purely phonemic model the labels for the last 2-3 phonemes.

6. While there are various common ways to z-score iEEG data, the way the authors do it is not standard and could introduce noise. For a given phoneme under analysis, the authors normalize the response to that individual phoneme based on only 100 ms of data that immediately precedes that phoneme. Two main concerns about this normalization technique:

(I) For iEEG 100 ms is too short to compute a reliable mean and standard deviation for baseline.

The authors should at least pool all the pre-target 100 ms segments and compute a global mean

and standard deviation for each electrode.

(II) There is no commonality between the baselines for different targets. Usually, baseline is taken as the pre-stimulus silence period, with the expectation that the response to different gaps of silence should be statistically similar. In this case however, the baseline is the response to the middle of a word or sentence which can vary wildly between epochs based on what came before. In such a case, normalizing by this variable period of response could introduce noise to the response.

Minor comments

- How are the phonemic labels aligned to the stimulus? Please add that information to the methods section.

- Figures 6 and 7 are never referenced in the text. Unclear:

Fig6: How are these "receptive fields" "reconstructed"

What is the reader supposed to understand from seeing them?

Fig7: Caption (correlation) doesn't match the figure text (explained variance)

Reviewer #2 (Remarks to the Author):

Mai, et al, "Acoustic and language-specific sources for phonemic abstraction from speech"

Summary: In this investigation, the authors point to a partial confound in previous investigations attempting to characterize the degree of abstraction in the neural response to speech, namely that abstract categories and features (e.g. phonetic features, phonemes, and syllables) tend to show a great deal of acoustic overlap as well. In this study, the authors capitalize on a more unusual situation where acoustic similarity and phonological similarity diverge to establish the degree to which neural encoding of speech abstracts away from acoustic similarity only, namely by looking at the way the brain processes tap sounds that have originated from either /t/ or /d/, and by looking at similarity among morphophonological variants of the past tense and plural endings in English. By doing so, the argument is made that the phoneme is a useful level of processing, a position I was already favorable disposed to.

This is a carefully executed study, and the discussion was sophisticated and clear. I had several methodological questions and one concern that I would like to see resolved before I feel confident about the author's primary assertion regarding the nature of the phoneme as a "real" level of neural representation.

Principally, I'd like to be reassured that the "phonemic" level of sensitivity shown here cannot possibly have come from higher level (e.g. phonotactic or lexical levels) of processing. Two points noted below that will help clarify this:

1. Pg 8 ln 212: The method the authors chose of estimating the likelihood of finding surface vs. underlying responses was clever—they elected to form 1,000 arbitrary pairings of two phonemes, then split one of those categories in an arbitrary half (A vs. Bx/By). This seems sensible, but given that all of the morphologically-related pairs are also one-feature phonetic changes (/s/ to /z/; /t/ to /d/) it might not be a truly fair comparison. I would recommend constraining this analysis to only those phoneme pairs that consist of a one-feature differences (and perhaps further constraining to those that only differ in voicing) to confirm that the distributions you see here would be unlikely to arise simply due to acoustic similarity. That is, a /s/ to /z/ assimilation pattern is more fairly compared to a /f/ vs /v/ pair that also differs in voicing (also a voicing change, but not deriving from the same underlying morpheme) than to a /f/ vs. /n/ comparison that differs on many more features. It would also be sensible to restrict the comparison set to the same phonological environment (e.g. intravocalic for taps; word-final for past-tense and plural). This is really quite key since these phonological changes occur in restricted phonological environments, and so it can be difficult to know if you are measuring differences in the neural response to the key token or the response to the surrounding phonology.

2. Section 3.4: One possibility is that what is characterized as a "phoneme-sensitive" response is really related to the listener's ability to predict/anticipate the likely input on the basis of prior context. That is, how can we be confident that the "phoneme-sensitive" response is really a phoneme response and not a lexical or semantic response? This interpretation is well-supported by the finding in Section 3.4 that phonemic information improves model fit for listening to a familiar language (English) but not an unfamiliar one (Catalan). This point emerges in the discussion, as well—in order to conclude that the following assumption is incorrect: "phonological knowledge, including knowledge of language-specific phonotactics, is ... epiphenomenal of lexical knowledge", you'd need to be very confident that you are really coding the phonological level, and not a bleed-over from properties of phonotactics and the lexicon. Unless the analysis constrains the "control" set to phonological environments that match the critical stimuli very well, that's hard to show.

Other issues:

As the supplementary materials note, coronal taps do tend to differ from one another acoustically depending on whether they were derived from /t/ and /d/--the authors compare the duration of the /t/-underlying and /d/-underlying taps and show that there is no difference. Yet as they note, there do tend to be consistent differences in the length of the preceding vowel (longer for underlying /d/ than /t/). I don't think that timelocking to the onset of the coronal tap nor baselining to the 100 msec prior the tap avoids this problem—the listener has already heard a longer vowel for /d/-derived tokens, so they may treat the same acoustic information (the tap) differently upon encountering the tap itself. As the authors are well aware, cues to phoneme identity are not located only within the duration of the phoneme itself—consider word-final stop voicing, where the best cue to the voicing of the stop is in the duration of the preceding vowel, not in the stop itself.

Page 5, ln 161-162: If I understand correctly, acoustic sites are those where the tap tokens (whether they originate from underlying /t/ or /d/) are distinct from the surface /t/ token—but why are these taps not also compared to the surface /d/ token? It is counterintuitive that surface /d/ wasn't part of the analysis scheme. Analogously, when it comes to the past-tense endings, why are both past-tense endings (e.g. “gapped”=/t/ and “bagged”=/d/) only compared to non-past-tense /d/ (e.g. “bad”) and not also to non-past-tense /t/ (e.g. “bat”)? In general this description of the underlying vs. surface comparison definitions could really benefit from a schematic.

Pg 5, ln 177-182: I expect I am missing some nuance of the phonological assumptions, specifically those motivating this statement: “The morphological surface similarity response groups plural [z] and non-plural [z] together to the exclusion of plural [s]”, but I don't understand why the plural formation analysis doesn't follow the same pattern as the tap pattern. That is, if surface -z and surface -s both originate from a plural form, they can be dissociated acoustically (but share an underlying representation).

(Figure 5): Why do all models containing spectrographic information perform so much better for listeners listening to Catalan than to English? Was the same (English) phoneme inventory used to code the Catalan data? How were equivalencies between English and Catalan phones mapped?

What do the authors make of the fact that the coronal tap alternation has many more “underlying-sensitive” electrodes, whereas the two morphological process have many more “surface-sensitive” electrodes?

Reviewer #3 (Remarks to the Author):

This paper provides neurophysiological evidence (from intra-cranial recordings in human patients with SEEG electrodes) for phonemes and morphemes. This paper is the most convincing demonstration to date of the neural reality of phonemes and morphemes. This is a remarkable finding because phonemes and morphemes are primarily abstract concepts put forward by linguistics analyses and their cognitive/neural relevance has often been questioned. Here, neural manifestations of these mental objects are captured during the processing of naturalistic, conversational speech.

The main empirical findings reported in the current paper are the following:

- * The response of some electrodes is driven by the surface phonetic representation, while the response of others is driven by the underlying phonemic/morphemic representation (Fig.2 table 1).

- * models that include both spectral information and phonemic/morphemic information fit the signal better than models that only include spectral information (or categorical information only).

- * phonemic labels better explain power in lower frequency bands while acoustic features better explain power in higher frequencies.

- * Phonemic labels have predictive power when the language is known to the participant (English), but not when it is unknown (Catalan).

This goes much beyond previous work (e.g. Mesgarani et al., 2014) that had identified responses to acoustic-phonetic features. The closest work is Di Liberto et al. (2015), who analyzed scalp EEG and showed the adding categorical phonemic information improved models' capacity to fit the signal. The current paper, using intracranial recordings, confirms and extends this previous work, as discussed in section 4.4.

In this type of studies, statistical analyses are crucial. The permutation analyses and the comparison between models' fit to English and Catalan speech convinced me that the results were not due to chance. Then, the discussion and supplementary materials document essentially addressed all the questions that I had when reading the results, e.g., about temporal and spatial aspects.

The paper is very well written, the analyses are a tour de force, the discussion is lucid. I do not have any major comment (only tiny suggestions listed below).

One remark is that the morphemes "plural" or "past tense" are confounded with the syntactic/semantics features "plural" or "past tense". In other words, for example, some electrodes could also respond to words which are "plural" even if they do not contain the plural morpheme. Maybe this could be worth discussing in section 4.2 (?)

Small remarks/suggestions

Page 5. line 159. "for a site to be considered an acoustic site, there must have existed at least one time windows with a significant test [...]". At this stage, you have not yet indicated the number and size of the time windows. This info is provided in the Methods section, p.25: 9 and 100 ms with an overlap of 50ms). I believe it would be a good idea to provide this information in the preceding sentence, line 158).

Fig.1 The caption could mention the total number of electrodes and the number of speech selective electrodes (for at least one band). This information is currently mentioned in in supp.mat. page 38). (I found that Figure S3 was very informative and wondered if it could not be a panel in Figure 1 (?))

p.9 Figure 3. Could you indicate in the caption the number of samples used to estimate the null distribution (1000?, I guess, from the sentence in line 214; or a different number in each case). This would help interpret the "0.0" proportions.

p.16, line 384 "to assess assessing" -> remove "assessing"

REVIEWER COMMENTS

Reviewer #1 (Remarks to the Author):

Summary

In this work, the authors try to distinguish between the acoustic and phonemic aspects of the neural response to speech, using the allophones of /t/ and /d/. Similarly, they try to distinguish the neural effects of morphological information of plural and past tense by comparing the word-final phonemes /s/ and /z/ for plural and /t/ and /d/ for past tense. They find that both phonemic and morphological information are represented in the neural signal, especially in the lower frequency band powers. The question the authors are addressing is very important and debated. However, there are major concerns about the methodological approach to answering this problem.

Major comments

1. The ratio of “speech selective” electrodes, as reported in Figure 1, is unreasonably high, especially considering that the coverage is mostly outside the auditory cortex. I think two incorrect assumptions cause this:

(I) The methods section (5.7 Significant Electrodes) is unclear on what component of the neural signal is being t-tested between speech and silence. Is it the raw signal? The power at a specific frequency band? While, for example, comparing the high-gamma power at an electrode site between speech and silence is meaningful, comparing the raw values is not as meaningful.

For each band, the difference in power is being tested. This has now been clarified in the text. It's possible that the number of speech responsive electrodes is higher than expected since the majority of our electrodes are in white matter and thus are likely recording activity that originates from a broader set of sources than typical cortical recordings.

(II) It is incorrect to apply the term “speech selective” when the selection method only guarantees that the response of the significant electrode sites differs between speech and silence. I suggest using the term “sound responsive” instead.

Thank you for this suggestion. We have changed the wording to “speech responsive” instead.

2. A primary advantage of intracranial EEG over noninvasive EEG or MEG is superior spatial resolution. Having a reasonably wide-spread coverage, I think the authors should have taken advantage of this fact and report the spatial implications of their findings. While I appreciate the section in the discussion dedicated to the differences between grey and white matter (4.3 Localization & Timing), the authors are in a position to address this in their results.

We agree that the spatial resolution of intracranial EEG (iEEG) is one of its primary advantages over noninvasive neuroimaging techniques, alongside its exceptional SNR. In this manuscript, we focus on characterizing the language-specific components of the iEEG response to speech.

In the originally submitted manuscript, in addition to discussion of localization and timing in Section 4.3, we also included a section in the Supplementary Materials providing an overview of the spatial distribution of results (as well as our reasoning for why we do not discuss these results more thoroughly in this particular manuscript). We believe that the spatial dynamics of the language-specific response to speech, especially given the broad coverage of observed speech responsive electrodes, deserve thorough investigation in their own right to be done correctly. We are continuing to look into those dynamics and may prepare a separate manuscript to focus on them.

3. Related to #2, when combining the responses of all electrodes to fit an encoding model (for example Figure 5), the high-frequency bands are at a disadvantage since they capture more local dynamics than low-frequency bands. This would make the LME model discard the local dynamics of a frequency band like high-gamma, which might include the phonological and morphological representations the authors are looking for. This also relates to the discussion raised by the authors in section 4.5 (Cross-participant variability in LME model fits).

The reviewer is asking whether we might be missing effects with the LME model in high-gamma, due to the differential contribution of local vs global dynamics over the different frequency components recorded at a given site. It is true, of course, that higher frequencies attenuate more quickly over distance than lower frequencies, and thus that the high-frequency components of the signal at each electrode reflect more local responses while lower-frequency components are integrated over larger anatomical space. Thus, all else being equal, any heterogeneity across sites (within or between participants) may more easily wash out LME effects in higher frequency bands than in lower frequency bands, since the latter are likely capturing a more global (ie. shared) response.

We don't disagree with this logic. To be clear, it is not our intent to imply that our failure to find strong phonemic effects in high frequency bands with the LME analysis should be taken as evidence that they do not exist. Indeed, several aspects of our results suggest they do. First, a significant number of sites do show sensitivity to underlying similarity in their high-gamma response, even after controlling for spatial correlations (Line 219-221). More importantly, for 3 of the 8 participants the s1p1 models provide the best fit to the data in either the gamma or high-gamma frequencies. Thus, the LME analysis can, and does, find phonemic effects in the higher frequencies. They just appear to be weaker than those in the lower frequencies.

In support of the idea that there are quantitative differences in the strength of the phonemic effects across bands, we note that separate LME's are run on each frequency band and each participant, so differing variance magnitudes across bands/subjects is not an issue. More importantly, the MNE analysis, which does not pool across sites but models the response of each site independently, reveals the same general pattern of contributions for different bands as the LME, and also clearly shows that all the bands carry phonemic information.

A paragraph has been added to the discussion section on cross-participant variability in LME model fits that addresses the possibility of band frequency itself contributing to the variability in LME model fits.

4. The usage of the GAIA autoencoder adds a confounding complexity to the problem without any significant payoff. First, a neural network autoencoder compressing 1 second of speech into a 128-dimensional vector can be learning complex “underlying” representations that are not simple acoustics, which importantly, invalidates the results of the LME analysis when comparing s1, p1 and s1p1 (Figure 4). Second, if the amount of compression desired is only 50% (256 -> 128), a linear dimensionality reduction method such as PCA could suffice. Alternatively, shorten the temporal window of the spectrogram from 1024 ms to 512 ms.

Every compression algorithm, neural network, PCA, or otherwise, is only capable of learning structure that already exists within the training set. Since the abstract underlying structure represented by the p1 feature set is fundamentally unrecoverable from speech acoustics alone (see “the lack of invariance problem”, e.g., Miller & Eimas, 1995, *Annual Review of Psychology*) the autoencoder-compressed spectrograms cannot contain phonological underlying representations.

Nevertheless, we agree that shortening the temporal window of the spectrogram also accomplishes the goal of reducing the dimensionality of the acoustic feature space and is generally more interpretable. We have rerun the analysis with a temporally-reduced (512ms window) and non-compressed feature set. As shown in Section 3.2 of the revised manuscript, the results are qualitatively similar to those of the models fit with GAIA-encoded feature sets, which have been moved to the Supplementary Materials.

5. Related to #4, a potentially complex representation of 1 second of speech spanning multiple words (Figure 4, left) is much more powerful/informative compared to a one-hot encoded phoneme vector. This makes the model comparison unfair. The equivalent purely phonemic model should have the phoneme labels for the last 1 second to be comparable. Since the dimensionality of such a model would be high, perhaps a similar autoencoder network could compress the phoneme labels of the 1-second window into a 128-dimensional vector. Alternatively, the authors can consider at least giving the purely phonemic model the labels for the last 2-3 phonemes.

We agree that the acoustic information provided to the models is both higher dimensional and more dense than the phonemic information provided to the models. This was done because most previous work in this area assumes that phoneme identity extraction can be reduced to preceding speech acoustics. Therefore, we take pains to provide highly informative acoustic features to avoid concerns that our phonemic features simply recapitulate information that would have been available in the speech acoustics if our acoustic features were more exhaustive. However, whether that higher dimensional representation is “more powerful/informative” with respect to explaining the neural response to speech is precisely the empirical question that this study addresses: Does the current phoneme identity explain aspects of the neural response that previous acoustic context does not? We believe the model features are appropriately selected to answer this question.

6. While there are various common ways to z-score iEEG data, the way the authors do it is not standard and could introduce noise. For a given phoneme under analysis, the authors normalize the response to that individual phoneme based on only 100 ms of data that immediately precedes that phoneme. Two main concerns about this normalization technique:

(I) For iEEG 100 ms is too short to compute a reliable mean and standard deviation for baseline. The authors should at least pool all the pre-target 100 ms segments and compute a global mean and standard deviation for each electrode.

(II) There is no commonality between the baselines for different targets. Usually, baseline is taken as the pre-stimulus silence period, with the expectation that the response to different gaps of silence should be statistically similar. In this case however, the baseline is the response to the middle of a word or sentence which can vary wildly between epochs based on what came before. In such a case, normalizing by this variable period of response could introduce noise to the response.

We have replaced the z-score baseline used in the original manuscript with a standard by-electrode mean subtraction baseline. Baselineing each response using a global mean for each electrode should result in a more reliable baseline. As the reviewer acknowledges, while use of a longer baseline would further reduce baseline noise in traditional trial-based experimental circumstances where baselines are highly uniform, it would increase noise for natural speech stimuli (where baselines are *a priori* more variable) by increasing the possible combinations of sounds present during the baseline period. For this reason, we continue using a 100ms baseline length, following Mesgarani et al. (2014).

Minor comments

- How are the phonemic labels aligned to the stimulus? Please add that information to the methods section.

The methods section in the submitted manuscript states that transcription, segmentation, and labeling procedures for passages from the Buckeye Corpus are described in Pitt (2007), and transcription, segmentation, and labeling of task instructions and content questions was performed by a phonetically-trained researcher at UC San Diego, using the protocols detailed in Pitt (2007). This passage has been revised to emphasize that segmentation and labeling are the act of providing time codes for the boundaries between phones and labels for the spans between boundaries, respectively.

- Figures 6 and 7 are never referenced in the text. Unclear:

Fig6: How are these "receptive fields" "reconstructed"?

What is the reader supposed to understand from seeing them?

References to and descriptions of the significance of Figures 6 and 7 have been added to the revised manuscript.

Fig7: Caption (correlation) doesn't match the figure text (explained variance)

Thank you for catching this. It has been corrected in the revised manuscript.

Reviewer #2 (Remarks to the Author):

Mai, et al, “Acoustic and language-specific sources for phonemic abstraction from speech”

Summary: In this investigation, the authors point to a partial confound in previous investigations attempting to characterize the degree of abstraction in the neural response to speech, namely that abstract categories and features (e.g. phonetic features, phonemes, and syllables) tend to show a great deal of acoustic overlap as well. In this study, the authors capitalize on a more unusual situation where acoustic similarity and phonological similarity diverge to establish the degree to which neural encoding of speech abstracts away from acoustic similarity only, namely by looking at the way the brain processes tap sounds that have originated from either /t/ or /d/, and by looking at similarity among morphophonological variants of the past tense and plural endings in English. By doing so, the argument is made that the phoneme is a useful level of processing, a position I was already favorably disposed to.

This is a carefully executed study, and the discussion was sophisticated and clear. I had several methodological questions and one concern that I would like to see resolved before I feel confident about the author’s primary assertion regarding the nature of the phoneme as a “real” level of neural representation.

Principally, I’d like to be reassured that the “phonemic” level of sensitivity shown here cannot possibly have come from higher level (e.g. phonotactic or lexical levels) of processing. Two points noted below that will help clarify this:

1. Pg 8 In 212: The method the authors chose of estimating the likelihood of finding surface vs. underlying responses was clever—they elected to form 1,000 arbitrary pairings of two phonemes, then split one of those categories in an arbitrary half (A vs. Bx/By). This seems sensible, but given that all of the morphologically-related pairs are also one-feature phonetic changes (/s/ to /z/; /t/ to /d/) it might not be a truly fair comparison. I would recommend constraining this analysis to only those phoneme pairs that consist of a one-feature differences (and perhaps further constraining to those that only differ in voicing) to confirm that the distributions you see here would be unlikely to arise simply due to acoustic similarity. That is, a /s/ to /z/ assimilation pattern is more fairly compared to a /f/ vs /v/ pair that also differs in voicing (also a voicing change, but not deriving from the same underlying morpheme) than to a /f/ vs. /n/ comparison that differs on many more features. It would also be sensible to restrict the comparison set to the same phonological environment (e.g. intravocalic for taps; word-final for past-tense and plural). This is really quite key since these phonological changes occur in restricted phonological environments, and so it can be difficult to know if you are measuring differences in the neural response to the key token or the response to the surrounding phonology.

The primary concern articulated here is that the large numbers of significant sites for the linguistically meaningful comparison could be driven by the phonological similarity of the comparisons (/t/ vs /d/; /s/ vs /z/). If this were true, we would expect the significance counts for random pairs (A,B) with smaller featural differences to be closer to the outer edge of the Monte Carlo distribution than the counts for random pairs with larger featural differences. However, this is not the case. In the revised manuscript, a figure has been added to the Supplementary Materials showing that the distribution of significant counts for random pairs with a single feature difference is not meaningfully distinct from the distribution of significant counts for random pairs with ten feature differences.

A secondary concern here is the consistency of the phonological environments surrounding the phonologically meaningful pairs vs. the randomly selected pairs. If the consistency in the phonological environment surrounding the phones participating in the tap, plural, and past tense alternations drove the number of significant sites observed for those comparisons, then we would expect comparisons of random phones in consistent phonological environments to accompany similarly large numbers of significant sites. To assess this, we performed the A, B_x, B_y analysis on 25 pseudo-random pairs of word-initial phones (i.e., all word-initial [z] vs. all word-initial [s]) and 25 pseudo-random pairs of word-final phones (i.e., all word-final [n] vs. all word-final [m]). Again, the distributions of randomly-chosen phones with consistent phonological environments falls well within the overall distribution of all randomly-chosen phones. In the revised manuscript, a figure has been added to the Supplementary Materials showing this.

2. Section 3.4: One possibility is that what is characterized as a “phoneme-sensitive” response is really related to the listener’s ability to predict/anticipate the likely input on the basis of prior context. That is, how can we be confident that the “phoneme-sensitive” response is really a phoneme response and not a lexical or semantic response? This interpretation is well-supported by the finding in Section 3.4 that phonemic information improves model fit for listening to a familiar language (English) but not an unfamiliar one (Catalan). This point emerges in the discussion, as well—in order to conclude that the following assumption is incorrect: “phonological knowledge, including knowledge of language-specific phonotactics, is ... epiphenomenal of lexical knowledge”, you’d need to be very confident that you are really coding the phonological level, and not a bleed-over from properties of phonotactics and the lexicon. Unless the analysis constrains the “control” set to phonological environments that match the critical stimuli very well, that’s hard to show.

The reviewer is asking how we can be confident that the “phoneme-sensitive” response is really a phoneme response and not a lexical or semantic response. We respectfully suggest that this question may not quite be framed accurately given what the paper shows, at least for Sections 3.2–3.4. With the LME and MNE models, we show that phonemic label information accounts for variance in the neural response that is not accounted for by our acoustic feature set. We do not show a phonemic response *per se*. For that reason, the more accurate concern is whether we think that our phonemic label features also implicitly contain lexical or semantic information. That is, could phonemic labels be capturing lexical or semantic aspects of the neural response?

Because we do not use models given sufficient sequential phonemic information to approximate wordlike information, Arbitrariness of Sign provides good confidence that the phonemic label features that we used do not contain lexical or semantic information. A paragraph addressing this concern has been added to the Discussion section.

Other issues:

As the supplementary materials note, coronal taps do tend to differ from one another acoustically depending on whether they were derived from /t/ and /d/--the authors compare the duration of the /t/-underlying and /d/-underlying taps and show that there is no difference. Yet as they note, there do tend to be consistent differences in the length of the preceding vowel (longer for underlying /d/ than /t/). I don't think that timelocking to the onset of the coronal tap nor baselining to the 100 msec prior the tap avoids this problem—the listener has already heard a longer vowel for /d/-derived tokens, so they may treat the same acoustic information (the tap) differently upon encountering the tap itself. As the authors are well aware, cues to phoneme identity are not located only within the duration of the phoneme itself—consider word-final stop voicing, where the best cue to the voicing of the stop is in the duration of the preceding vowel, not in the stop itself.

The surrounding phonological environment is essential to all accounts of phonological alternation. From Panini to the present, phonology has been based on the premise that the patterns of sounds in the languages of the world are conditioned on their context. As an undoubtedly familiar example, *The Sound Pattern of English* (Chomsky & Halle, 1968) introduces regular rewrite rules of the form A → B / C_D (in words, “the underlying sound A surfaces acoustically as sound B in the context between C and D”). The fact that the surrounding phonological environment provides cues to the underlying phonological identity of sounds is not a wrinkle that can be worked around. It is the nature of phonology itself.

For this reason, it is not concerning that having heard the preceding context, “[the listener] may treat the same acoustic information (the tap) differently upon encountering the tap itself.” We believe this is precisely what phonological processing requires. That very process *is* phoneme identity extraction.

The point of the baselining is to ensure that what is observed in the time window of interest is not merely the sum of what came before it. The surrounding context certainly matters; what is interesting (and what this paper shows) is that the response to acoustically similar sounds (i.e., various taps) diverges in a way that is consistent with the extraction of phonemic information and is irreducible to prior acoustic context alone.

Page 5, In 161-162: If I understand correctly, acoustic sites are those where the tap tokens (whether they originate from underlying /t/ or /d/) are distinct from the surface /t/ token—but why are these taps not also compared to the surface /d/ token? It is counterintuitive that surface /d/ wasn't part of the analysis scheme. Analogously, when it comes to the past-tense endings, why are both past-tense endings (e.g. “gapped”=/t/ and “bagged”=/d/) only compared to non-past-tense /d/ (e.g. “bad”) and not also to non-past-tense /t/ (e.g. “bat”)?) In general this

description of the underlying vs. surface comparison definitions could really benefit from a schematic.

A priori, it was decided to investigate only one “anchor” for each of the comparisons: underlying /t/ for the tap comparison, underlying /d/ for the past tense comparison, and underlying /z/ for the plural comparison. For the tap comparison, underlying /t/ was chosen because surface [t]/[tʰ] are generally more acoustically distinct from taps than surface [d], and we wanted it to be particularly unlikely that underlying sites for the tap comparison (those grouping [t]/[tʰ] with /t/-taps) could be explained away as another kind of acoustic similarity. For the plural and past tense, /d/ and /z/ anchors were chosen because they are typically considered to be the underlying forms of the regular past and plural morphemes, respectively.

Nevertheless, it is expected that the /d/ anchor for the tap comparison, the /t/ anchor for the past tense comparison, and the /s/ anchor for the plural comparison should also yield numbers of surface and underlying responses greater than those of the null distribution. For the revision, we have created a figure that adds the values for these three additional comparison types/anchors to Figure 3. As expected, all counts lie outside the 95% bound of the null distribution. We think that introducing the idea of “anchors” will make the main text needlessly more complex, so we have placed this figure in the Supplementary Materials and explained the three additional comparisons there.

We have also added a schematic illustration of the three comparisons and different anchor types to the Supplementary Materials.

Pg 5, In 177-182: I expect I am missing some nuance of the phonological assumptions, specifically those motivating this statement: “The morphological surface similarity response groups plural [z] and non-plural [z] together to the exclusion of plural [s]”, but I don’t understand why the plural formation analysis doesn’t follow the same pattern as the tap pattern. That is, if surface -z and surface -s both originate from a plural form, they can be dissociated acoustically (but share an underlying representation).

For the morphological comparison, the prevalence of both “surface” sites and “underlying” sites was identified. As the reviewer notes, “The morphological surface similarity response groups plural [z] and non-plural [z] together to the exclusion of plural [s]”. This is how the “surface” sites are defined for both the plural and past comparisons. They then raise the possibility that plural surface [z] and plural surface [s] may dissociate acoustically but share an underlying representation. This is precisely how “underlying” sites are defined in the next paragraph, where we state that “morphological underlying sites were those for which there was at least one time window indicating a significant difference between word-final non-plural /z/ and plural /s/ tokens and between word-final non-plural /z/ and plural /z/ tokens but no significant difference between plural /z/ and plural /s/ tokens” (lines 192-6).

In the cited lines (177-182), the point being made is that plural and non-plural [z] are both acoustically similar (= [z]) and phonologically similar (= /z/), so surface sites as defined for the

morphological comparison are in this sense indistinguishable from both surface (phonetic) sites and underlying (phonological) sites as defined for the tap comparison. That is, the cover terms “surface sites” and “underlying sites” have slightly different interpretations for the phonological comparison (tap) compared to the morphophonological comparisons (plural and past tense).

(Figure 5): Why do all models containing spectrographic information perform so much better for listeners listening to Catalan than to English? Was the same (English) phoneme inventory used to code the Catalan data? How were equivalencies between English and Catalan phones mapped?

Thanks for bringing this to our attention. We’ve added text to the discussion explaining why we think the spectrographic models for Catalan outperform those for English. We think it’s most likely an effect of attention, since the participants were asked to listen carefully for a recognizable word in the Catalan speech, but were not asked to listen so carefully to the English speech.

As for the Catalan labeling scheme, the phone labels used for the Catalan speech were drawn from the phonetic transcriptions provided with the corpus of speech. We’ve added a phrase to the methods section clarifying this.

What do the authors make of the fact that the coronal tap alternation has many more “underlying-sensitive” electrodes, whereas the two morphological processes have many more “surface-sensitive” electrodes?

As mentioned above, the definition of surface sites for the morphophonological comparisons encompasses the definitions of both surface and underlying sites defined for the phonological comparison. For this reason, we think it is possible that the larger number of underlying sites for the phonological comparison and of surface sites for the morphophonological comparisons may be related. This is mentioned in the caption of Table 1.

Reviewer #3 (Remarks to the Author):

This paper provides neurophysiological evidence (from intra-cranial recordings in human patients with SEEG electrodes) for phonemes and morphemes. This paper is the most convincing demonstration to date of the neural reality of phonemes and morphemes. This is a remarkable finding because phonemes and morphemes are primarily abstract concepts put forward by linguistics analyses and their cognitive/neural relevance has often been questioned. Here, neural manifestations of these mental objects are captured during the processing of naturalistic, conversational speech.

The main empirical findings reported in the current paper are the following:

* The response of some electrodes is driven by the surface phonetic representation, while the response of others is driven by the underlying phonemic/morphemic representation (Fig.2 table 1).

* models that include both spectral information and phonemic/morphemic information fit the signal better than models that only include spectral information (or categorical information only).

* phonemic labels better explain power in lower frequency bands while acoustic features better explain power in higher frequencies.

* Phonemic labels have predictive power when the language is known to the participant (English), but not when it is unknown (Catalan).

This goes much beyond previous work (e.g. Mesgarani et al., 2014) that had identified responses to acoustic-phonetic features. The closest work is Di Liberto et al. (2015), who analyzed scalp EEG and showed the adding categorical phonemic information improved models' capacity to fit the signal. The current paper, using intracranial recordings, confirms and extends this previous work, as discussed in section 4.4.

In this type of studies, statistical analyses are crucial. The permutation analyses and the comparison between models' fit to English and Catalan speech convinced me that the results were not due to chance. Then, the discussion and supplementary materials document essentially addressed all the questions that I had when reading the results, e.g., about temporal and spatial aspects.

The paper is very well written, the analyses are a tour de force, the discussion is lucid. I do not have any major comment (only tiny suggestions listed below).

One remark is that the morphemes "plural" or "past tense" are confounded with the syntactic/semantics features "plural" or "past tense". In other words, for example, some electrodes could also respond to words which are "plural" even if they do not contain the plural morpheme. Maybe this could be worth discussing in section 4.2 (?)

Yes, we agree there are quite possibly sites that would respond to [plural] regardless of its exponence, and it would take a follow-up study to assess whether the underlying sites that we observe for the plural and past tense are sensitive only to the regular plural and past tense (i.e. exponent specific) or to "plural-ness" more generally. We've added a bit of text to this effect in the Discussion section, as suggested.

Small remarks/suggestions

Page 5. line 159. "for a site to be considered an acoustic site, there must have existed at least one time windows with a significant test [...]". At this stage, you have not yet indicated the number and size of the time windows. This info is provided in the Methods section, p.25: 9 and 100 ms with an overlap of 50ms). I believe it would be a good idea to provide this information in the preceding sentence, line 158).

Thank you for this suggestion. We've added this information into the main text to make this clearer, as suggested.

Fig.1 The caption could mention the total number of electrodes and the number of speech selective electrodes (for at least one band). This information is currently mentioned in supp.mat. page 38). (I found that Figure S3 was very informative and wondered if it could not be a panel in Figure 1 (??))

I've tested adding Figure S3 as a panel in Figure 1, but I think it makes the overall figure too large, so for now I have left the figures as they are. I am willing to add Figure S3 to Figure 1 if the Editor thinks it would also be useful and appropriate.

In addition to being mentioned in the Methods and Supplemental Materials sections, the average number of speech responsive electrodes per band is also provided at the beginning of the fifth paragraph of Section 3.1.

p.9 Figure 3. Could you indicate in the caption the number of samples used to estimate the null distribution (1000?, I guess, from the sentence in line 214; or a different number in each case). This would help interpret the "0.0" proportions.

Good point. We've now added that the null distribution has 1000 samples for each band to the caption for Figure 3.

p.16, line 384 "to assess assessing" -> remove "assessing"

Thank you, fixed.

REVIEWERS' COMMENTS

Reviewer #1 (Remarks to the Author):

The authors have added further analysis to address my main concerns. I have no further comments.

Reviewer #2 (Remarks to the Author):

I appreciate the authors' clarifications in this revised manuscript, and my prior concerns have been addressed. Only one minor concern emerged--see below.

Minor:

Supplementary Figure 2: The grey distribution is very difficult to see--

Reviewer #3 (Remarks to the Author):

My evaluation of the first version was quite positive and my (minor) comments have been addressed by the authors in the new version. Therefore I am happy to recommend this manuscript for publication.

REVIEWERS' COMMENTS

Reviewer #1

The authors have added further analysis to address my main concerns. I have no further comments.

Thank you for all your comments throughout this process!

Reviewer #2

I appreciate the authors' clarifications in this revised manuscript, and my prior concerns have been addressed. Only one minor concern emerged--see below.

Minor:

Supplementary Figure 2: The grey distribution is very difficult to see--

The null distribution has been darkened in the revised submission. Thank you also for all your comments throughout this process!

Reviewer #3

My evaluation of the first version was quite positive and my (minor) comments have been addressed by the authors in the new version. Therefore I am happy to recommend this manuscript for publication.

Thank you for all your comments throughout this process!